# Spatiotemporal formation of glands in plants is modulated by MYB-like transcription factors

Jiang Chang[1,2], Shurong Wu[1,2], Ting You[1,2], Jianfeng Wang[1], Bingjing Sun[1], Bojun Xu[1], Xiaochun Xu[1], Yaping Zhang[1] & Shuang Wu [1] ✉

About one third of vascular plants develop glandular trichomes, which produce defensive compounds that repel herbivores and act as a natural biofactory for important pharmaceuticals such as artemisinin and cannabinoids. However, only a few regulators of glandular structures have been characterized so far. Here we have identified two closely-related MYB-like genes that redundantly inhibit the formation of glandular cells in tomatoes, and they are named as *GLAND CELL REPRESSOR* (*GCR*) 1 and 2. The *GCR* genes highly express in the apical cells of tomato trichomes, with expression gradually diminishing as the cells transition into glands. The spatiotemporal expression of *GCR* genes is coordinated by a two-step inhibition process mediated by SlTOE1B and GCRs. Furthermore, we demonstrate that the *GCR* genes act by suppressing *Leafless* (*LFS*), a gene that promotes gland formation. Intriguingly, homologous *GCR* genes from tobacco and petunia also inhibit gland formation, suggesting that the GCR-mediated repression mechanism likely represents a conserved regulatory pathway for glands across different plant species.

Plant hairs, also known as trichomes, cover the surface of most terrestrial plants, and are often used as one of the key traits in plant taxonomy. In nature, trichomes exhibit an enormous diversity of morphology and size in different species. Trichomes can be unicellular, such as *Arabidopsis* trichomes and cotton fibres, but in most other plant species, they are multicellular[1–3]. In about 30% of all vascular plant species, trichomes form glandular cells at the top where a large amount of specialised metabolites are produced[4,5]. Many of the gland-produced compounds act to protect plants such as acyl sugars, nicotine and various terpenoids for defensive purposes[6–9]. Some of them such as artemisinin, cannabinoid and essential oils, are of importance pharmaceutical and industrial value[10–12]. Insights into the molecular mechanisms behind gland formation may provide strategies for genetic modification of plants with enhanced stress resistance and bio-production of valuable metabolites[13].

Our current knowledge of trichome development is largely derived from *Arabidopsis* trichome, a type of non-glandular unicellular trichome. Important genes that regulate trichome initiation, patterning and branching, as well as the associated and cellular processes have been well characterised using the *Arabidopsis* trichome system[14–18]. However, multicellular trichomes in plant species other than *Arabidopsis* appear to have different regulatory mechanisms, as GL1, the key trichome regulator identified in Arabidopsis, has been shown not to be involved in trichome development in tobacco[19]. In tomato (*Solanum lycopersicum*), there are seven different types of trichomes with different morphologies and functions, which are named as type I–VII[20]. Among these trichomes, type I–V appear to be long with a small single glandular head or without, and are thus named as digital trichomes (DT), whereas type VI and type VII are short and form a multicellular glandular head which are

[1]College of Horticulture, FAFU-UCR Joint Center for Horticultural Biology and Metabolomics, Fujian Agriculture and Forestry University, Fuzhou, China.
[2]These authors contributed equally: Jiang Chang, Shurong Wu, Ting You. ✉e-mail: wus@fafu.edu.cn

named as peltate trichomes (PT) (Supplementary Fig. 1a). Interestingly, the trait of trichomes appeared to have changed during tomato domestication, as many wild species (*S. habrochaites* and *S. pennellii*) form very dense glandular trichome. In contrast, these seven types of trichomes are all be found in modern cultivated tomatoes, but vary in different tissues during development[20,21].

A member of HD-ZIP family named *Woolly* (*Wo*) is a key regulator of the formation of both glandular and non-glandular trichomes in tomato, and presumably in many other plant species that form multi-cellular trichomes[22,23]. Wo plays a dose-dependent role in the tri-chome differentiation in tomato, with high levels of Wo promoting the DT formation through activating MX1 and Wox3B regulatory module, and low Wo levels favoring PT fate by activating *Leafless* (*LFS*), an AP2 gene[24,25]. In addition, many other positive regulators have also been reported, including C2H2 zinc finger proteins *Hair* and *SlHair-2*, as well as the other member of HD-Zip IV, *Lanata* (*Ln*)[26–28]. On the other hand, several negative regulators have also been identified. An E3-ligase MTR1 has been shown to decrease Wo protein levels during trichome development[23,24]. Another member of HD-Zip IV, SlHD8, was found to restrict the trichome length in tomato[29]. Plant hormone, gibberellin, jasmonic acid and auxin have been reported to positively affect trichome initiation in tomato[30–32]. However, these factors appear to function mostly at an early stage of trichome development, and generally affect many types of tri-chomes. It remains unclear how the glandular cells, one of the major structures of plant trichomes, are formed. In this study, we identify two novel repressors of gland formation and elucidate the mechan-ism underlying the spatiotemporal formation of glandular trichomes in tomato.

## Results

### MYB-like transcription factors act as key repressors of gland formation

To find regulators associated with gland formation, we screened for transcription factors with significantly higher expression in glandular trichomes than in the leaves or stems with all trichomes removed. *S. pennellii* mostly forms only type IV glandular trichomes (Supplementary Fig. 1b), making it an ideal system for the transcriptome analysis. Among the glandular trichome-expressed transcription factors derived from the transcriptome of *S. pennellii*, we identified 22 transcription factors that are also highly expressed in the previous glandular cell tran-scriptomes of *S. lycopersicum* LA4024 and *S. habrochaites* LA1777[33] ('Methods'; Supplementary Fig. 1c, d; Supplementary Data 1). These identified transcription factors include the previously reported *EOT1*, *HD8* and *LFS*, which are related to trichome development and meta-bolism, suggesting the validity of the analyses[9,24,29]. Using the CRISPR-Cas9 technique, we knocked out all previously uncharacterised genes on this list. Based on the trichome phenotypes of single and double-mutants, we identified two closely related (81% homology) transcription factors (Solyc02g076670 and Solyc03g006150), both of which contain a MYB-like domain with a conserved SHLQMY and EAR motif (Supple-mentary Fig. 1e, f).

Scanning electron microscopy (SEM) showed that mutations in these two genes dramatically promoted gland formation in tomato trichomes, and we thus named them as *GLAND CELL REPRESSOR* (*GCR*) 1 and 2 (Fig. 1a–d). However, the overall plant growth was not affected, suggesting the function of these two *MYB*-like genes is restricted to glandular trichomes (Supplementary Fig. 2a–d). To quantitatively assess the glandular trichome phenotype, we divided all trichomes

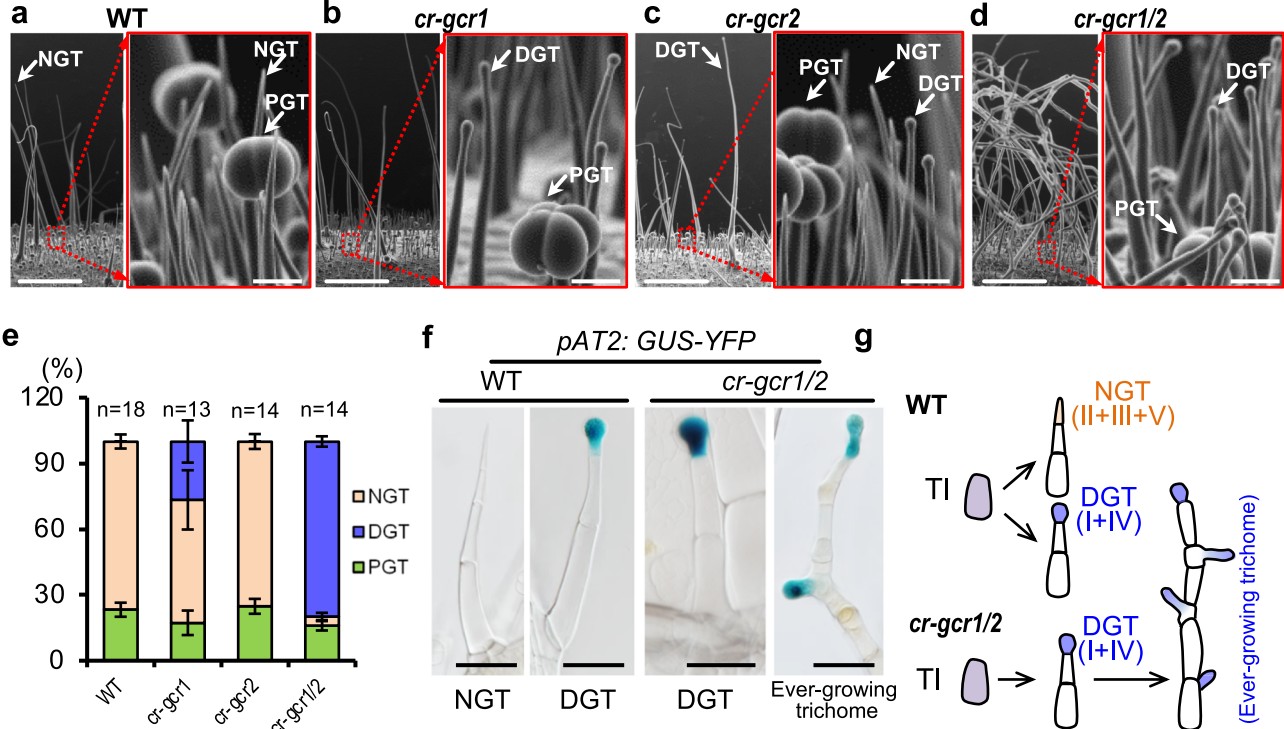

**Fig. 1 | Knockout of *GCR1/2* repress gland formation. a–d** Trichome phenotype of *gcr1* (*cr-gcr1*), *gcr2* (*cr-gcr2*) and *gcr1/2* double mutant (*cr-gcr1/2*). Right panels with red frame are the close-up views of the trichomes in the left images. Bar: 1 mm (left images); 50 μm (right images). NGT: non-glandular trichoems; DGT: digitate glandular trichomes; PGT: peltate trichomes. **e** Quantification of the trichomes on the adult stems. The *Y*-axis represents the proportion of three categories of tri-chomes in the total number of trichomes. *n* represents the number of SEM images used for the quantification. For each line, at least 13 different SEM images from three to five individual plants were used for trichome quantification. Data are shown as mean ± SD. *p*-values were obtained by unpaired two-sided *t*-test and presented in the Supplementary Table 2. **f** Identification of glandular cells by GUS staining. Marker of glandular cells (*pAT2*: GUS-YFP) is introduced into *gcr1/2* double mutant (*cr-gcr1/2*). Bar: 100 μm. **g** Schematic diagram of over-proliferated gland phenotype in *gcr1/2* double mutants. Purple cells represent trichome initials (TI); pale yellow cell is non-glandular cell; blue cells are glandular cells.

into three categories: NGT (non-glandular trichomes including type II, III and V), DGT (digital glandular trichomes including type I and IV) and PGT (peltate glandular trichomes including type VI and VII) (Supplementary Fig. 1a). Our quantification showed that the *gcr1* single mutant had about 26% of NGT converted into DGT, whereas the *gcr2* single mutant had no significant phenotype. In contrast, the *gcr1/2* double mutant had a more dramatic phenotype with 96% of trichomes producing glandular structures (Fig. 1a–e). Furthermore, in WT, glandular cells usually form in the apical region, which represents the determinate differentiation of the trichome. However, *gcr1/2* double mutants exhibited an ever-growing trichome phenotype, with glands overproliferating in cells located in the middle part of the trichome (Fig. 1f, g, Supplementary Fig. 3). Interestingly, the short PGT also produced supernumerary glandular heads in the double mutants (Supplementary Fig. 2e), suggesting both transcription factors are gland repressors. The quantification showed that the long digital trichomes (type I/II) of WT had an average length of 3 mm, with 8 cells within a trichome cell file, whereas this type of trichomes reached up to 5 mm and contained about 30 cells within a trichome cell file in *cr-gcr1/2* mutants (Supplementary Fig. 4). As a result, trichomes in double mutants became significantly elongated (Fig. 1d–g, Supplementary Fig. 2d).

Consistent with their functions, GCR1 expressed in tomato trichomes, with the expression being stronger in the apical cells of the

non-glandular trichomes (NGT) than in the glandular cells of DGT and PGT (Fig. 2a, b, Supplementary Fig. 5). To further know whether the abolished expression of GCR1/2 in apical cells is essential for glandular cell formation, we forced the expression of GCR1/2 in trichome apical cells by the *MTR1* promoter. *MTR1* promoter was previously reported to be active specifically in all trichomes, usually with a higher activity in apical cells (Supplementary Fig. 6a)[24]. The ectopic expression of GCR1/2 entirely inhibited gland formation (Fig. 2c, d, Supplementary Fig. 6b–e). Such inhibitory role could also be verified in cultivated tomato Alisa Craig (AC), *S. pimpine* LA1589 and *S. pennellii* LA0716 (Supplementary Fig. 7). As most *Solanaceae* species form glandular trichomes, we tested the function of *GCR* homologues by forcing the expression of *NbGCR* and *PeGCR* cloned from *N. benthamiana* and *Petunia* respectively under the *MTR*1 promoter. The result showed that these *GCR* genes play a conserved role in inhibiting glandular cell formation (Fig. 2c, d).

### SlTOE1B antagonises GCR1/2 in gland formation

To elucidate the mechanism that by which GCR1/2 inhibits gland formation, we performed a yeast two-hybrid screen for GCR1 interactors in a yeast cDNA library that was constructed with tomato trichomes and shoot apical tissues. An AP2/ERF transcription factor (Solyc04g049800) was highly enriched in the screened clones (Supplementary Fig. 8a,

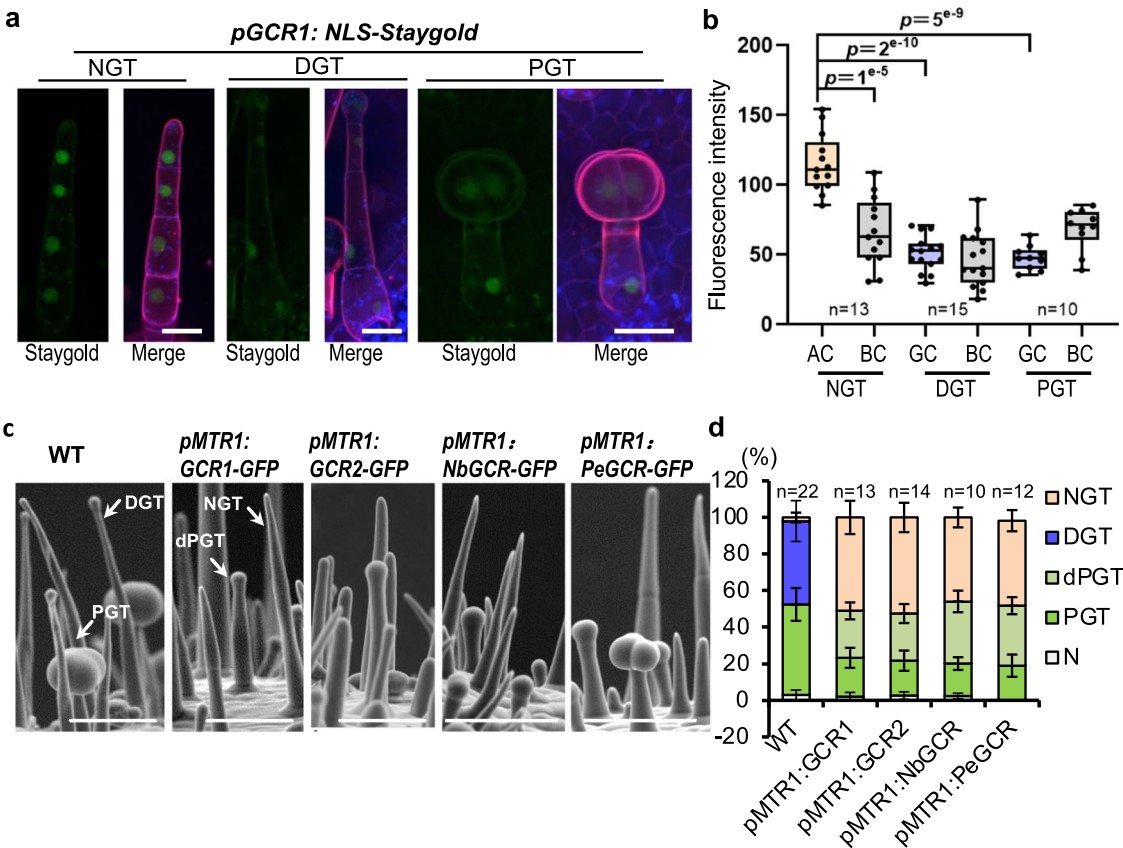

**Fig. 2 | GCR1/2 repress gland formation. a** Expression pattern of *GCR1*. The promoter of *GCR1* fused with NLS-Staygold and stable transformed into wild type tomato. Leaves of ten-day-old seedlings were used for analysis. Propidium Iodide (PIE) staining was used to show the cell edge. Chloroplast auto-fluorescence is shown in blue. All images were collected under the same condition and processed by maximum intensity projection of z-stacks. Bar: 20 μm. **b** GCR1 expression is quantified by fluorescence intensity measurement of z-stacked images of *pGCR1:NLS-Staygold*. Box plots show maximum, minimum, first and third quartiles, median (line). N represents the number of trichomes used for the quantification. At least 15 trichomes from two lines were used to quantify. BC: Basal cell; AC: Apical

cell. *p*-values obtained by unpaired *t*-test. ns: no significant. **c** Expression of GCR1/2 and its homologues in trichome cells by *MTR1* promoter represses the gland formation. Pictures show trichomes on the hypocotyl. dPGT: defective peltate trichomes. Bar: 100 μm. **d** Quantification of trichomes on juvenile stems. The *Y*-axis represents the proportion of three categories of trichomes in the total number of trichomes. n represents the number of SEM images used for the quantification. For each line, at least ten different SEM images from four to eight individual plants were used for trichome quantification. Data are shown as mean ± SD. *p*-values were obtained by unpaired two-sided *t*-test and presented in the Supplementary Table 3. N: defective trichomes or developing trichomes.

Supplementary Table 1). As the homologue of this gene in *Arabidopsis* is *TOE1*, we named it as *SlTOE1B* (Supplementary Fig. 8b). Yeast two-hybrid, BiFC, pull-down and Co-IP all validated the interaction between GCR1/2 and SlTOE1B in vitro and in vivo, and the C-terminus of GCR1/2 mainly interacted with SlTOE1B (Fig. 3a, b, Supplementary Fig. 8c, d, Supplementary Fig. 9). Visualisation of *pSlTOE1B: NLS-Venus* revealed that *SlTOE1B* was highly expressed in the glandular cells of DGT and PGT, and had no expression in non-glandular trichomes (Fig. 4a, b).

Next, we forced the expression of *SlTOE1B* using *MTR1* promoter. In *Arabidopsis*, *TOE* is the target of miR172, and *SlTOE1B* coding region also carries the target site of *miR172*. We thus introduced silent mutations to the *miR172* target site in *SlTOE1B* to generate miR172-resistant form of *SlTOE1B*, which we named as *rSlTOE1B* (Fig. 4c). Different levels of the *rSlTOE1B* expression caused distinct degrees of the increase in glandular cells (Fig. 4d, e, Supplementary Fig. 10). When *SlTOE1B* expression was increased 2-fold, all non-glandular trichomes (types II, III and V) disappeared and were replaced by digital glandular trichomes (type I and IV). When *SlTOE1B* expression was increased ten-fold, glandular cells of both DGT and PGT increased (Supplementary Fig. 10). To further verify the SlTOE1B function in gland formation, we generated the *cr-sltoe1b* mutant by CRISPR/Cas9. Consistent with the forced expression result, *cr-sltoe1b* mutant showed the reduced DGT

and increased NGT (Fig. 4f, g, Supplementary Fig. 11a). However, not all glandular trichomes were abolished in *cr-sltoe1b* mutant, suggesting the existence of functional redundancy. This opposite phenotype to that caused by GCR1/2 expression suggests the potential antagonism between SlTOE1B and GCR1/2.

## GCR/SlTOE/TPL2 complex suppress *GCR1/2* expression

To further dissect the relationship between GCR and SlTOE, we crossed *pMTR1: GCR1-GFP* and *pMTR1:rSlTOE1B-GFP*, and the result showed that the expression of *pMTR1: GCR1-GFP* repressed the gland formation induced by *pMTR1: rSlTOE1B-GFP*, indicating that *GCR1* is genetically epistatic to *SlTOE1B* (Fig. 5a, b). The trichome phenotype of the *gcr1 sltoe1b* double mutant was the same as that of *cr-gcr1*, supporting that *GCR1* is indeed epistatic to *SlTOE1B* (Supplementary Fig. 11b, c). Furthermore, we found that the expression of *GCR1/2* was strongly inhibited in *pMTR1: rSlTOE1B-GFP*, but interestingly was significantly enhanced in *cr-gcr1/2* (Supplementary Fig. 12a, b). We therefore speculated that both SlTOE1B and GCR1/2 might be involved in the repression of *GCR1/2* expression. To test this, we quantified the endogenous expression levels of *GCR1* and *GCR2*, and our results showed that increased *GCR1/2* expression indeed suppressed the endogenous expression of *GCR1* and *GCR2* (Supplementary Fig. 12c).

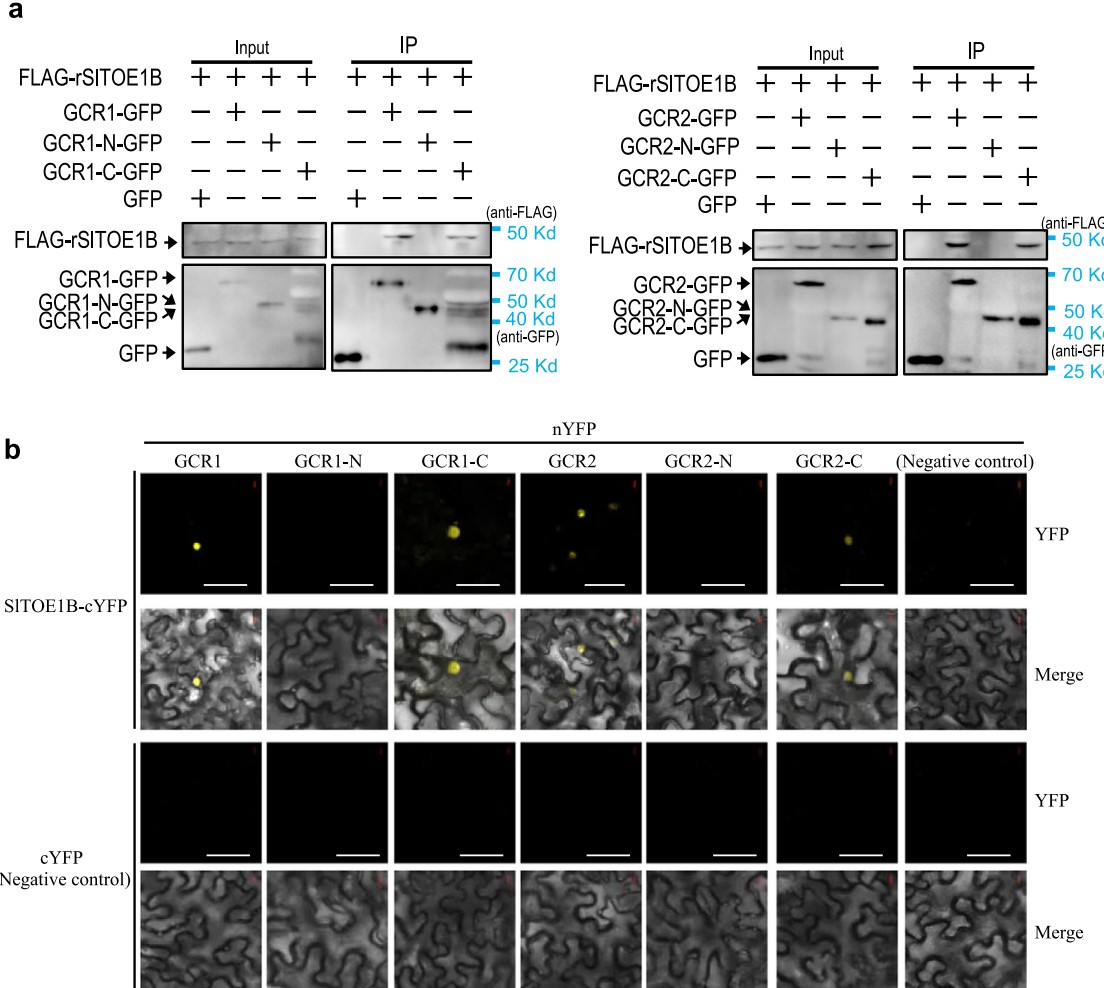

**Fig. 3 | GCR1/2 interact with SlTOE1B in vivo. a** Co-immunoprecipitation (CoIP) assay shows GCR1/2 and GCR1/2-C could interact with SlTOE1B in vivo. GCR1, N-terminal domain of GCR1 (GCR1-N), C-terminal domain of GCR1 (GCR1-C), GCR2, GCR2-N and GCR2-C were fused with GFP. SlTOE1B fused with FLAG. GFP (left lane, negative control) or GFP fused with the target protein (other lanes) were used as bait to bind to the GFP beads. Immunoprecipitated proteins were detected by anti-

FLAG. **b** Results of bimolecular fluorescence complementation (BiFC) assays in *N. benthamiana* leaf epidermal cells show GCR1/2 interacted with SlTOE1B through its C-terminal domain. GCR1/2 and their fragments fused with C-terminal domain of YFP (nYFP). SlTOE1B fused with N-terminal domain of YFP (cYFP). Three biological repeats were performed for each interaction. Bar: 50 μm.

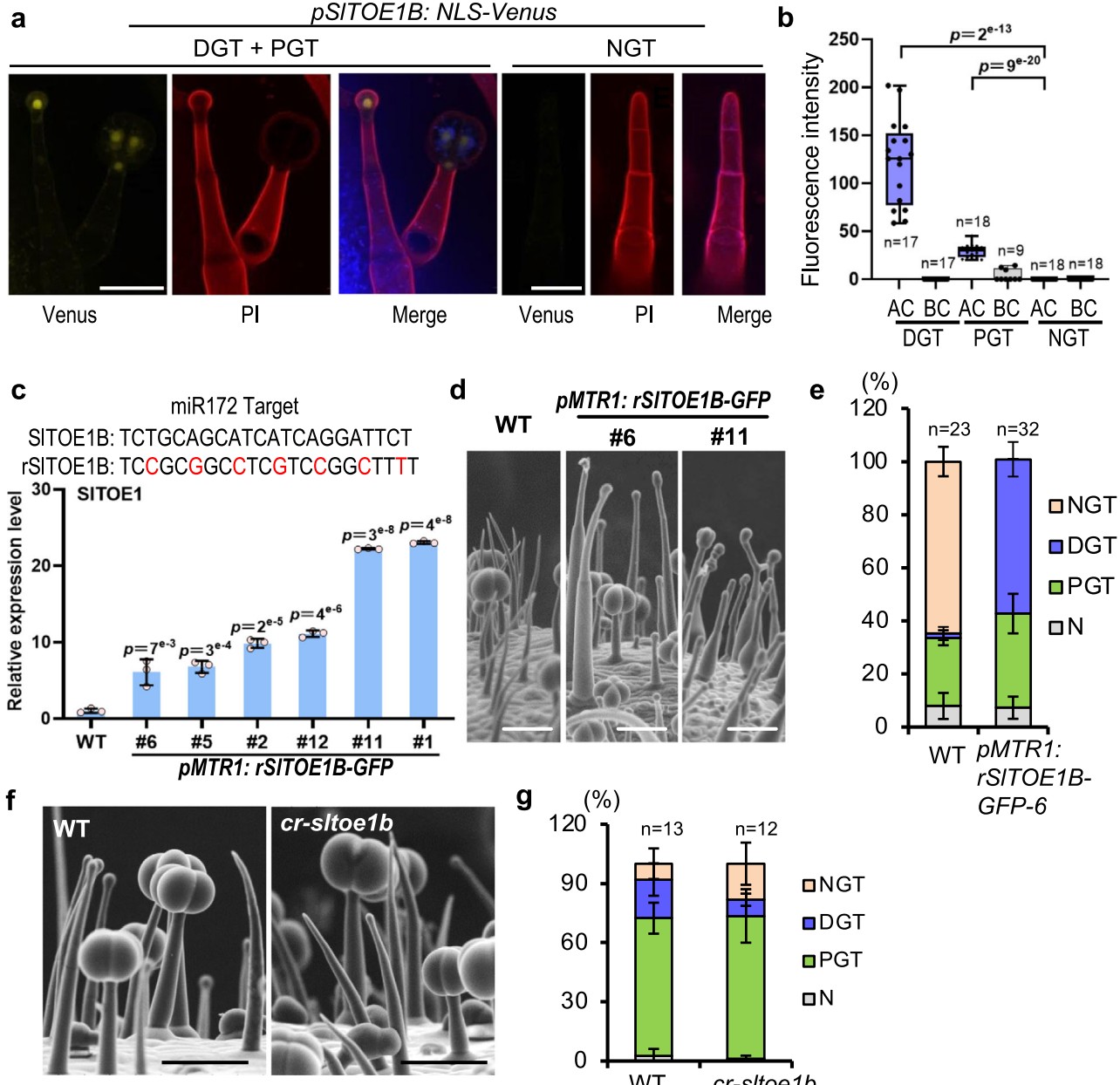

**Fig. 4 | SlTOE1B promotes gland formation. a** Expression pattern of SlTOE1B. Leaves of 10-day-old seedlings of *pSlTOE1B:NLS-Venus* transgenic plants were analysed. PIE staining was used to show the cell edge. Chloroplast auto-fluorescence is shown in blue. All images were collected under the same condition and processed by maximum intensity projection of z-stacks. Bar: 50 μm. **b** SlTOE1B expression was quantified by fluorescence intensity measurement of z-stacked images of *pSlTOE1B:NLS-Venus*. Box plots show maximum, minimum, first and third quartiles, median (line). *n* represents the number of trichomes used for the quantification. At least two lines were quantified. **c** Expression of SlTOE1B in trichome cells by *MTR1* promoter promotes gland formation. Nucleotide sequences show the miR172 target. Silent mutations are introduced to the miR172 target site in SlTOE1B to generate miR172-resistant form of SlTOE1B (marked as rSlTOE1B). Histogram shows the relative expression level of SlTOE1B in six *pMTR1: rSlTOE1B* stable transgenic lines. Data are shown as mean ± SD (*n* = 3 biological replicates). **d** SEM images show the trichome phenotype of line 6 and line 11. Bar: 100 μm. **e** Quantification of trichomes on the adult stems. The *Y*-axis represents the proportion of three categories of trichomes in the total number of trichomes. *n* represents the number of SEM images used for the quantification. 23 different SEM images from four individual WT plants and 32 different SEM images from five individual plants of *pMTR1: rSlTOE1B* were used for trichome quantification. Data are shown as mean ± SD. N: defective trichomes or developing trichomes. **f** SEM images show the trichome phenotype of the juvenile stems of *cr-sltoe1b*. Bar: 100 μm. **g** Quantification of trichomes on the juvenile stems. The *Y*-axis represents the proportion of three categories of trichomes in the total number of trichomes. n represents the number of SEM images used for the quantification. 13 different SEM images from seven individual WT plants and 12 different SEM images from eight individual plants of *cr-sltoe1b* were used for trichome quantification. Data are shown as mean ± SD. *p*-values were obtained by unpaired two-sided *t*-test in (**b**, **c**, **e** and **g**) and exact *p*-values of (**e** and **g**) are presented in the Supplementary Tables 4 and 5, respectively.

Furthermore, we employed luciferase reporter assay in tomato protoplasts of both WT and *cr-gcr1/2*. We found that GCR1/2 was indeed able to inhibit its own expression. In contrast, SlTOE1B was able to enhance GCR1/2 self-inhibition (Fig. 5c, Supplementary Fig. 12d, e).

According to the JASPER database (https://jaspar.elixir.no), GCR homologous genes in *Arabidopsis* (AT2G38300 and AT2G40260) can potentially bind to cis-element containing HHHATTCYHHH or HHWHATTCYHHH. Interestingly, we identified 13 such sequences in

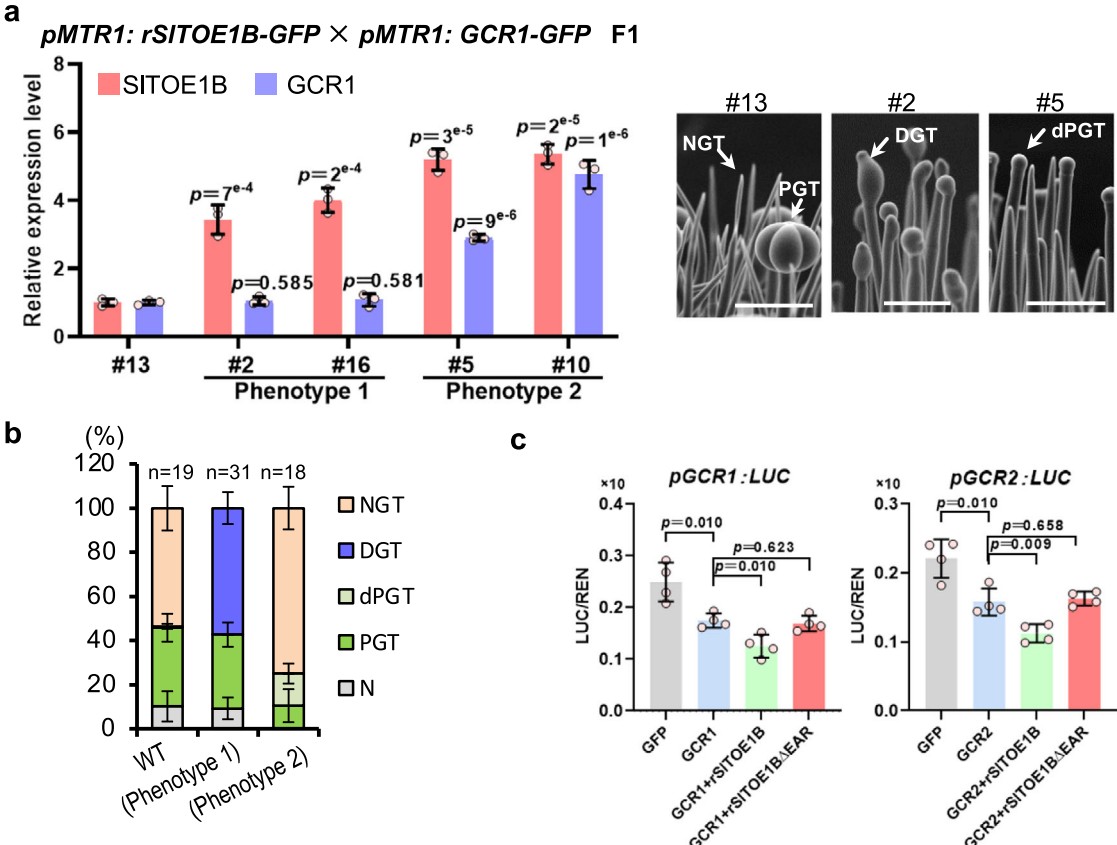

**Fig. 5 | GCR1/2 inhibit their own expression through SlTOE1B. a** GCR1 is epistatic to SlTOE1B in the gland formation pathway. The relative expression level of SlTOE1B and GCR1 were detected in the F1 plants obtained by crossing *pMTR1: rSlTOE1B-GFP* and *pMTR1: GCR1-GFP*. The *X*-axis represents different individual plants (biological replicates). Data are shown as mean ± SD ($n = 3$ technical replicates). Exact *p*-values were calculated by unpaired two-sided *t*-test. Trichome phenotype of Line 2 (#2) and Line 6 (#6) is the same as *pMTR1: rSlTOE1B-GFP* plants (Phenotype 1) and trichome phenotype of Line 4 (#4) and Line 5 (#5) is the same as *pMTR1: GCR1-GFP* plants (Phenotype 2). SEM images show trichomes of line 13 (#13), line 2 (#2) and line 5 (#5). Bar: 100 μm. **b** Quantification of trichomes on the juvenile stems of F1 plants shown in a. The *Y*-axis represents the proportion of three categories of trichomes in the total number of trichomes. *n* represents the number of SEM images used for the quantification. For each line, at least 19 different SEM images from three to five individual plants were used for trichome quantification. Data are shown as mean ± SD. *p*-values were calculated by unpaired two-sided *t*-test and are presented in the Supplementary Table 6. **c** LUC reporter assays in *cr-gcr1/2* protoplasts. GCR1 and GCR2 repress the transcriptional activity of their own promoter, and rSlTOE1B is able to enhance the repression. Deleting EAR motifs of rSlTOE1B abolish the repression (rSlTOE1BΔEAR). *p*-values were calculated by unpaired two-sided *t*-test. $n = 4$ biological replicates.

the 3 kb upstream of GCR1 and ten in the 3 kb upstream of GCR2. We then conducted Y1H screen to test whether GCR1/2 can bind to these predicted motifs (Supplementary Fig. 13). Next, we performed Y1H to validate that both GCR1 and GCR2 interacted with their own promoters (Fig. 6a). Furthermore, we performed biotin-IP using His-GCR1/2 fusion protein which was purified from *Escherichia coli* strain BL21 (DE3). We also conducted ChIP-qPCR using p35S:GCR1-GFP transgenic plants. The results of both biotin-IP and ChIP-qPCR were consistent with Y1H (Fig. 6b, c). These results suggest that *GCR1/2* could maintain a homoeostasis of its own expression in the early stage of trichome development, while SlTOE1B forms a complex with GCR1/2 to enhance such inhibition to abolish *GCR1/2* expression when the apical cells differentiate into glands.

In *Arabidopsis*, TOE1 acts as a transcriptional repressor by recruiting Topless (TPL) via its EAR motif[34–36]. We thus analysed the total of 6 *TPL* genes that are identified in the tomato genome. In Y2H assay, we detected a strong interaction of SlTOE1B with SlTPL2 and a weak interaction with SlTPL4 (Supplementary Fig. 14a). The SlTOE1B protein contains two AP2 domains, a C-terminal EAR motif (LDLNL) and an N-terminal EAR-LIKE motif (EARL, LDLNN). Further pull-down, Y2H and BiFC revealed that both the EAR and EAR-like motifs of SlTOE1B were involved in the physical interaction between SlTOE1B and SlTPL2 (Supplementary Fig. 14b–e). In luciferase reporter assay,

deletion of these EAR motifs abolished the inhibitory effect of SlTOE1B on the transcriptional activity of *pGCR1/2* (Fig. 5c). Based on these results, we hypothesise that GCR1/2 recruit SlTOE1B/TPL2 to form a repressive complex to maintain a lower expression level of *GCR1/2*, which favours the differentiation into glands. In non-glandular cells, where SlTOE1B is not expressed, GCR1/2 self-inhibition is weakened and the high level of GCR1/2 blocks the gland formation.

In modern cultivars tomato, DGT formation occurs mainly at juvenile stages when miR156-SPL/miR172-TOE modules control the developmental transition[21]. Interestingly, the ectopic expression of miR156B by either the 35S or *MTR1* promoter promoted gland formation (Supplementary Fig. 15a–e). In *pMTR1:miR156B* lines, we found that *SlTOE1B* expression was dramatically increased, whereas *GCR1/2* expression was strongly repressed (Supplementary Fig. 15f). In addition, the cross between *pMTR1:GCR1-GFP* and *35S:miR156B* impaired miR156B-promoted gland formation, suggesting a function of *GCR1/2* downstream of *miR156* in gland regulation (Supplementary Fig. 16a, b).

**GCR1/2 inhibits gland formation via repressing *LFS* expression**
To further identify genes downstream of GCR1/2, we isolated the trichomes of *pMTR1:GCR1-GFP* and *cr-gcr1/2* for RNA-seq. We chose the genes that were down-regulated in *pMTR1:GCR1-GFP* and up-regulated in *cr-gcr1/2*. The result was further intersected with the

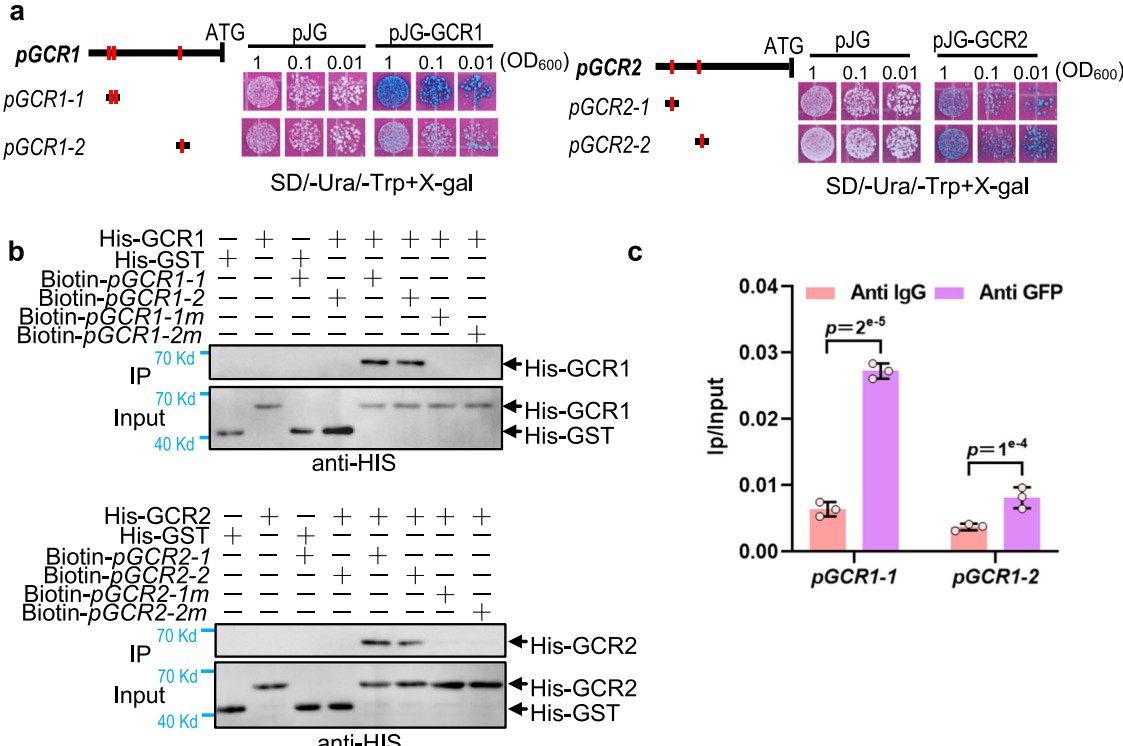

**Fig. 6 | GCR1/2 directly bind to their own promoters. a** GCR1 and GCR2 bind to their own promoters by Y1H. GCR1 binds to its promoter fragments *pGCR1-1* and *pGCR1-2* containing motif 1 and 3 (referred to Supplementary Fig. 13). GCR2 interacts with its own promoter fragments *pGCR2-1* and *pGCR2-2* containing motif 9 and 10 (referred to Supplementary Fig. 13). The bold black lines represent the promoter sequence of GCR1 and GCR2. The red vertical lines indicate motifs. Three cell dilutions for each interaction were presented. **b** Biotin-labelled DNA IP assays show GCR1 interacts with *pGCR1-1* and *pGCR1-2* and GCR2 interacts with *pGCR2-1* and

*pGCR2-2*. Mutation motifs of promoter fragments (*pGCR1-1m, pGCR1-2m, pGCR2-1m, pGCR2-2m*) disrupt the interaction between GCR1/2 and its own promoter fragments. Promoter fragments and mutant fragments labelled with biotin were used as bait and incubated with streptavidin-agarose beads. His-GST protein was used as negative control. Immunoprecipitated proteins were detected by anti-His. **c** The ChIP-qPCR analysis shows the interaction between GCR1 and the promoter of *GCR1*. Data are shown as mean ± SD (*n* = 3 biological replicates). *p*-values were calculated by unpaired two-sided *t*-test.

22 transcription factors mentioned above (Fig. 7a, Supplementary Fig. 1d, Supplementary Data 2). Two transcription factors including HD9 and LFS were obtained via the analyses (Fig. 7a, b). *Leafless* (*LFS*) encodes an AP2 transcription factor that has been reported to promote the fate of peltate trichomes in tomato[24]. As *lfs* null mutant has strong cotyledon and leaf phenotypes[25], we chose a heterozygote (line #1) and a double heterozygote mutant (line #3) for the phenotypic analysis. In line with the previous report, the glands of DGT were almost all lost in *cr-lfs* mutants (Supplementary Fig. 17a–c). Interestingly, the gland phenotypes varied in *cr-lfs* mutants, which seemed to be associated with the LFS levels. In some heterozygous lines (#1), only the glands of DGT were inhibited, whereas all glands in both PGT and DGT were suppressed in homozygous lines (#3, Supplementary Fig. 17a–c).

Consistent with the RNA-seq data, qRT-PCR confirmed that *LFS* expression was inhibited in *pMTR1:GCR1-GFP*, but enhanced in *cr-gcr1/2* (Supplementary Fig. 17d, e). This regulation of LFS by GCR1/2 was also validated by luciferase reporter assay (Fig. 7c). Interestingly, the *LFS* promoter also contains GCR1/2 preferred binding motifs (Supplementary Fig. 13). Y1H, biotin-IP and ChIP-qPCR showed that GCR1/2 were able to bind directly to the *LFS* promoter (Fig. 7d–g).

Although GCR1/2 are not typical MYB transcription factors, mutations in the MYB domain, as shown by the *cr-gcr1/2*-20 line in which a 63 bp deletion in the MYB domain occurred, disrupting their inhibitory role on trichome glands. Consistent with this, mutations of three conserved amino acids within the MYB domain of GCR1/2 abolished their repression of *LFS* expression in both Y1H and luciferase reporter assays (Fig. 7c–e). Furthermore, we crossed *cr-gcr1/2* and *cr-lfs* and we found no gland formation in *gcr1/gcr2/lfs* triple mutants

(Fig. 8a, b). Taken together, we conclude that GCR1/2 inhibit gland formation by repressing LFS expression.

## GCR1/2 represses *LFS* expression via TPL2

Since GCR1/2 also carry an EAR motif, it is possible that GCR1/2 play an inhibitory role by recruiting TPL. In support of this hypothesis, both GCR1 and GCR2 exhibited a strong interaction with SlTPL2 and a weak interaction with SlTPL4 (Supplementary Fig. 18a). Further pull-down, Co-IP and BiFC experiments confirmed such interactions (Fig. 9a, b, Supplementary Fig. 18b, c). Furthermore, GCR1/2 interacted with TPL only through their C-terminal domain that contains the EAR motif. Once EAR motif was removed, the interaction was attenuated, suggesting that the EAR motif is essential for GCR/TPL interaction (Fig. 9a, b, Supplementary Fig. 18b, c). In *cr-gcr1/2* protoplasts, the addition of SlTPL2 substantially enhanced the inhibition of LFS by GCR1/2. In contrast, GCR1/2ΔEAR, in which the EAR motif was deleted, lost the ability to repress LFS expression (Fig. 9c). Tomato *tpl2* mutants, generated by CRISPR-Cas9, also displayed the phenotype of increased glands (mostly in PT) and branched trichomes (mostly in DT) (Fig. 9d, Supplementary Fig. 19).

In summary, trichome is an age-dependent feature in tomato. In juvenile plants, trichomes appeared to be all glandular types including DGT and PGT. In contrast, trichomes formed in adult plants include both NGT and PGT, suggesting phase-related signals such as miR156 may be involved in trichome regulation. However, when both GCR1 and two were knocked out, supernumerary glands formed throughout the all developmental stages. In contrast, forced expression of *GCR1/2* genes in trichome apical cells inhibited the gland formation (Fig. 10a). In WT, the distribution of glandular and

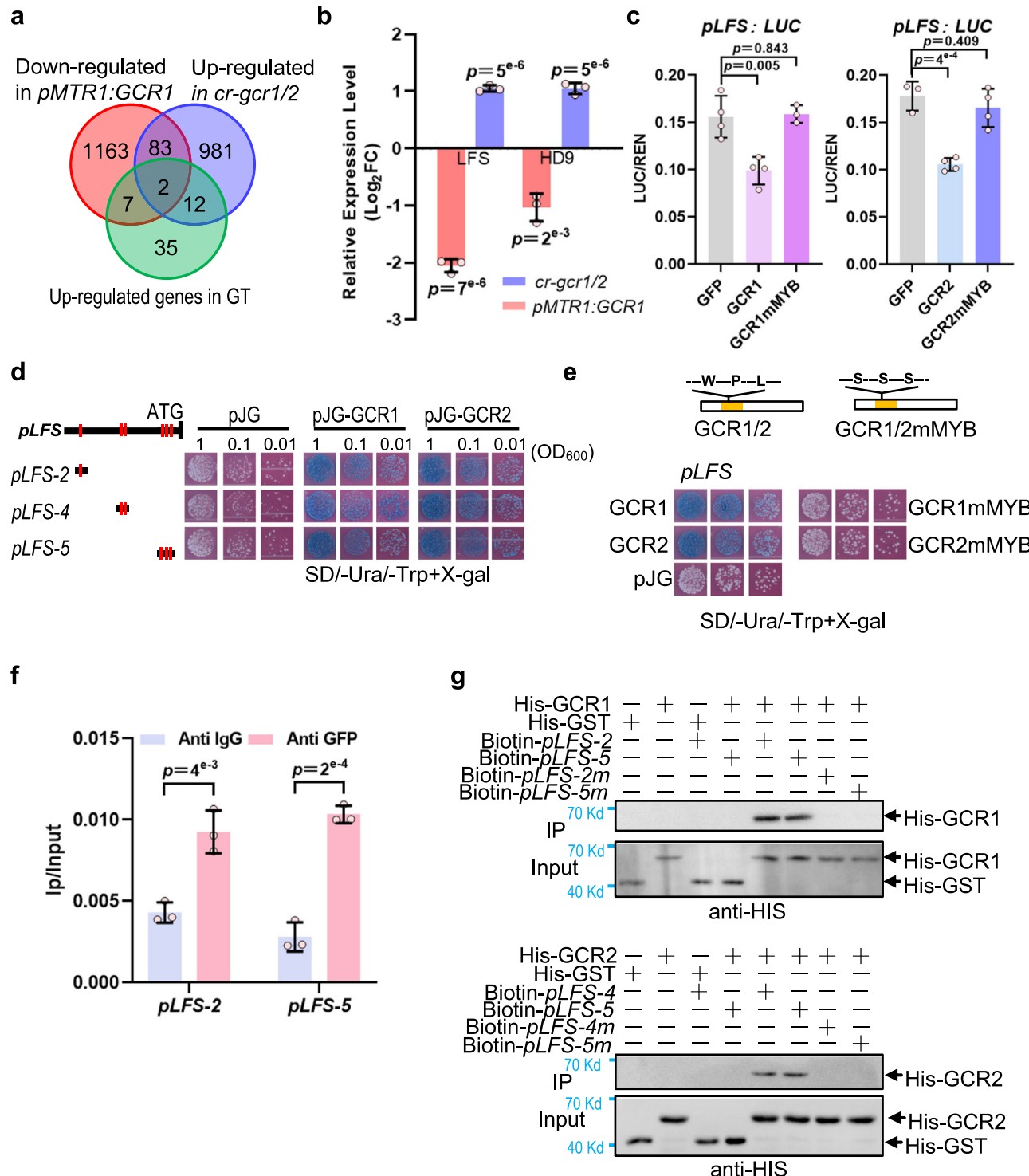

non-glandular trichomes appears to rely on the spatiotemporal expression of *GCR1/2*. Both genes were highly expressed in apical cells of non-glandular trichomes, but low expression in the apical cells of glandular trichomes. This spatiotemporal expression of *GCR1/2* probably results from a combination of self-inhibition and SlTOE1B induced inhibition (this is further discussed in the Discussion section). *SlTOE1B* expression was initiated at the apical cells when the gland cells start to form, and interaction with SlTOE1B appeared to dramatically enhance GCR self-inhibition. Once the GCR expression was removed, LFS was activated, which leads to the formation of gland heads. The mechanism behind gland formation also highlights the key role of TPL-mediated repression (Fig. 10b).

## Discussion

After the initiation in the epidermis, the construction of tomato trichomes is achieved through the spatial arrangement of cell division and cell differentiation[37–39]. The glands often begin with the specific fate decisions in trichome apical cells. Therefore, the spatiotemporal expression of key regulators is essential for the establishment of such a developmental pattern. Glandular cells are formed in about 20–30% of vascular plants[4,5], but in some species, including tomato, both glandular and non-glandular trichomes are formed simultaneously[20]. This suggests that there must be a specific switch in the apical cells to decide either the glandular fate or the subsequent cell division into a conical structure. Since division starts with the trichome initiation, cell

**Fig. 7 | GCR1/2 directly repress the expression of *LFS*. a** Venn diagram showing the overlap of up-regulated transcriptional factors in gland (referred to Supplementary Fig. 1), down-regulated genes in *pMTR1: GCR1* and up-regulated genes in *cr-gcr1/2*. **b** The relative expression level of *LFS* and *HD9* in *pMTR1: GCR1* and *cr-gcr1/2* based on the transcriptome data. **c** LUC assay. GCR1 and GCR2 inhibited the transcriptional activity of LFS promoter and mutation of the MYB domain in GCR1 and GCR2 (GCR1-mMYB, GCR2-mMYB) disrupted the inhibition in tobacco protoplast. **d** Y1H shows GCR1 and GCR2 bind to the promoter fragments of *LFS*. The bold black lines represent the promoter sequence of *LFS*. The red vertical lines indicate motifs (referred to Supplementary Fig. 13). GCR1 binds to *LFS* promoter fragments *pLFS-2*, *pLFS-4* and *pLFS-5* containing motif 1, 6, 7, 8 and 9 (referred to Supplementary Fig. 13). GCR2 interacts with LFS promoter fragments *pLFS-4* and *pLFS-5*. Three cell dilutions for each interaction were presented. **e** Y1H shows mutation of conserved amino acids in MYB domain of GCR1 and GCR2 abolish the interaction between

GCR1/2 and LFS promoter. The black boxes represent the amino acid sequence of GCR1 and GCR2. The yellow boxes show the MYB domains. Three conserved amino acids (Trp, W; Pro, P; Leu, L) in the MYB domain of GCR1 and GCR2 are replaced with serine (S) to construct GCR1mMYB and GCR1mMYB. SD/−2: -Ura/-Trp. Three cell dilutions for each interaction were presented. **f** The ChIP-qPCR analysis shows the interaction between GCR1 and the promoter of LFS. **g** Biotin-labelled DNA IP assays show GCR1 interacts with *pLFS-2* and *pLFS-5* and GCR2 interacts with *pLFS-4* and *pLFS-5*. Mutation motifs of promoter fragments (*pLFS-2m, pLFS-4m, pLFS-5m*) disrupt the interaction between GCR1/2 and its own promoter fragments. Promoter fragments and mutant fragments labelled with biotin were used as bait and incubated with streptavidin-agarose beads. His-GST protein was used as negative control. Immunoprecipitated proteins were detected by anti-His. Data are shown as mean ± SD (*n* = 3 (**b**), 3 - 4 (**c**) and 3 (**g**) biological replicates). *p*-values were calculated by unpaired two-sided *t*-test.

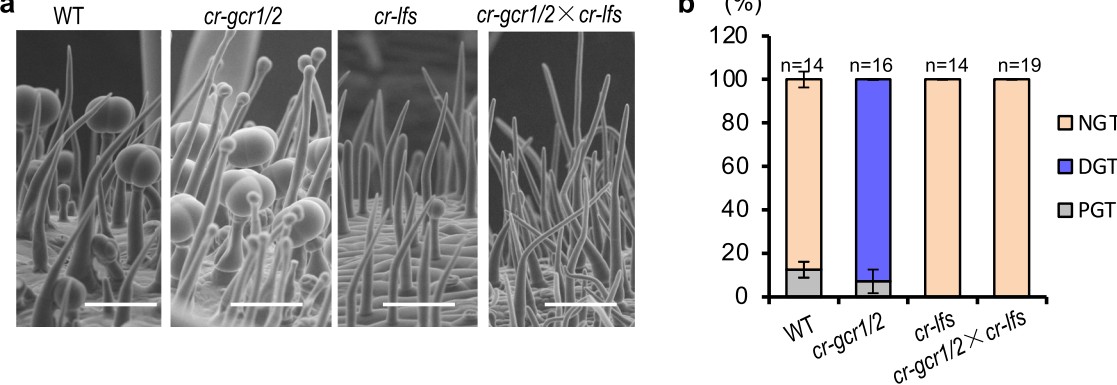

**Fig. 8 | GCR1/2 repress the gland formation by inhibiting LFS. a** SEM images showing no gland formation in *gcr1/gcr2/lfs* triple mutants obtained by crossing *cr-gcr1/2* and *cr-lfs*. Bar: 100 μm. **b** Quantification of trichomes on the petioles of WT, *cr-gcr1/2*, *cr-lfs* and *gcr1/gcr2/lfs* triple mutants (*cr-gcr1/2×cr-lfs*). The *Y*-axis represents the proportion of three categories of trichomes in the total number of

trichomes. n represents the number of SEM images used for the quantification. For each line, at least 14 different SEM images from three to six individual plants were used for trichome quantification. Data are shown as mean ± SD. *p*-values were calculated by unpaired two-sided *t*-test and are presented in Supplementary Table 7.

division seems to be a default process, while a gland-promoting factor is spatiotemporally activated in the apical cell. However, our results reveal that the real regulation is much more complex than this assumption, involving the repressor and feedback loops.

In this study, we identified two closely related glandular cell repressors (GCR1/2) that appear to function in a dose-dependent manner and whose levels need to be tightly controlled. Several mechanisms could contribute to the regulation of GCR1/2 levels during trichome development. First, GCR1/2 are capable of self-inhibition, and such repression is presumably mediated by their EAR motifs. EAR-containing transcription factors have been reported to recruit TPL to play a repressive role in the regulation of downstream genes in a wide range of processes including hormone signalling, meristem maintenance, organogenesis and reproduction[34–36,40]. In these processes, TPL was found to further recruit histone deacetylase HAD to deactivate the downstream transcription[34–36,40,41]. In tomato, SlERF.F12 can bind to TPL2, which further recruit SlHDA1 and SlHDA3, to form a tripartite complex to suppress the fruit ripening[40]. Here we found the disruption of *SlTPL2* only partially recovered the gland phenotype, suggesting that additional TPL genes may be involved. Secondly, SlTOE1B can physically interact with GCR1/2 to significantly enhance the self-inhibition of *GCR1/2*. Interestingly, a homologue of SlTOE1B has recently been reported to act as a transcriptional repressor in the inflorescence branching of tomato[42]. In our study, we found SlTOE1B alone appears to be capable of inhibiting *GCR1/2* expression. This inhibition might be important in the glandular cells when GCR1/2 abundance drops to an extremely low level. In this study, we have not found the direct binding site of SlTOE1B in the putative *GCR1/2*

promoters. In *Arabidopsis*, AtTOE1 binds to the 3' UTR of *AtGL1*, forming a chromatin looping to repress *AtGL1* expression[43]. It is possible that SlTOE1B may be associated with the region outside of the examined GCR1/2 promoters (3000 bp upstream of the genes). Third, based on the spatiotemporal expression pattern of *GCR1/2* and *SlTOE1B* during trichome development, we propose a putative working model: *GCR1/2* expression is initiated at the early stage of trichome development, and is gradually increased and restricted in the apical cells of the non-glandular trichomes. Upon differentiation into glandular cells in the apical region, the primed apical cells begin to express *SlTOE1B* and the expression was gradually enhanced in the apical glandular cells (Supplementary Fig. 22). In this way, the rising GCR1/2 expression needs to be turned down quickly, and the weakly expressed SlTOE1B needs a more efficient way to do this. At this stage, binding of SlTOE1B to GCR1 provides a more efficient way to reverse the rising GCR1 expression. Once this trend is reversed and the GCR1 levels are lower than SlTOE1B levels, SlTOE1B may be able to directly inhibit GCR1 expression and eventually eliminate it from the developing glandular cells. Finally, GCR1/2 and their downstream target, LFS, may form a negative feedback loop that has been shown in many biological systems to maintain the homoeostasis of the signalling or regulation. Dual luciferase reporter assay showed that LFS was able to activate the expression of SlTOE1B (Supplementary Fig. 20), supporting the feedback activation of SlTOE1B by LFS. This complex regulation may provide high plasticity in response to both developmental programmes and environmental stimuli.

After the initiation in the epidermis, the construction of glandular trichomes is achieved through the spatial arrangement of cell division

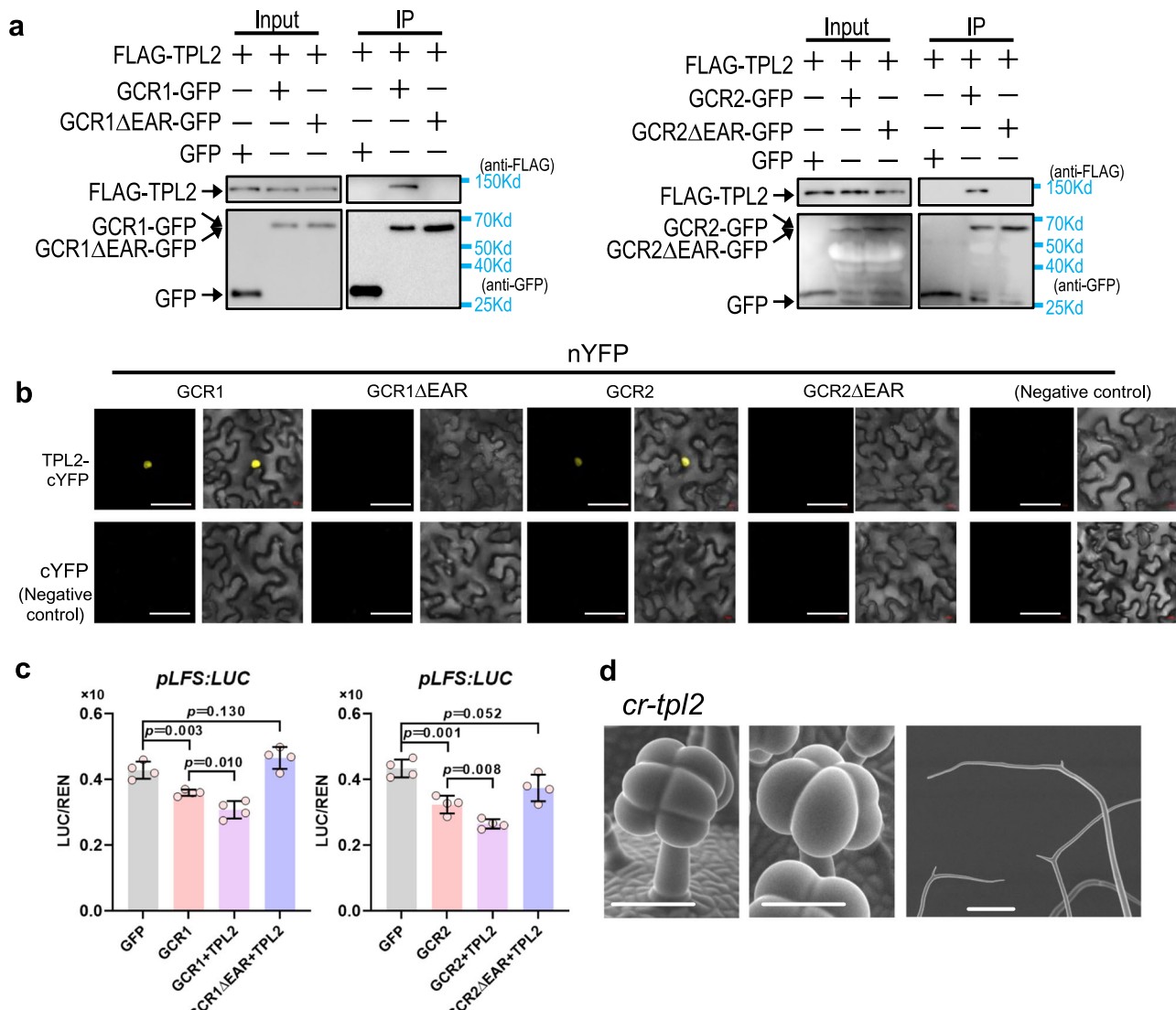

**Fig. 9 | GCR1/2 inhibits LFS expression by recruiting TPL2. a** CoIP shows GCR1 and GCR2 interact with TPL2 by EAR motif. GCR1/2-GFP deleted the EAR motif are marked as GCR1ΔEAR-GFP and GCR2ΔEAR-GFP. GFP (left lane, negative control) or GFP fused with the target protein (other lanes) were used as bait to bind to the GFP beads. Immunoprecipitated proteins were detected by anti-FLAG. **b** Results of BiFC assays in *N. benthamiana* leaf epidermal cells show GCR1/2 interacted with TPL2 and deletion of EAR motif disrupts the interaction. GCR1/2 and GCR1/2 with deletion of EAR fused with nYFP. TPL2 fused with cYFP. Bar: 50 μm. **c** LUC assay showed GCR1/2 and TPL2 repress the transcriptional activity of LFS promoter and deletion of EAR motif in the GCR1/2 (GCR1ΔEAR) abolish the inhibition. *p*-values were calculated by unpaired two-sided *t*-test. *n* = 5 biological replicates. **d** Knockout of TPL2 promotes gland formation of some PT and branched trichomes. Bar: 200 μm.

and cell differentiation[37–39]. Disruption of GCR1/2 produced trichomes with more than three times the number of cells compared to WT. In contrast, forced expression of GCR1/2 in apical cells resulted in defective trichomes with only 1–2 cells, indicating that GCR1/2 also repress cell division in trichomes. In the unicellular *Arabidopsis* trichome, cell division is not required once the trichome is initiated. Subsequent morphogenesis involves repeated endoreduplication, which requires the function of SIM, a CDK inhibitor[18,44]. Interestingly, loss-of-function of SIM and its homologous gene in *Arabidopsis* results in cell division in the unicellular trichomes[44,45]. However, the morphogenesis of multicellular trichomes relies on the spatially coordinated cell division and endoreduplication. In tomato, gain-of-function of *Woolly* may cause the increased cell division in the multicellular trichomes[22,46]. However, further research is needed to understand how GCR1/2 integrates these division-associated factors to regulate cell division in tomato trichomes.

Glandular trichomes are important for plant defence and specialised metabolite production, which have received increasing attention

in recent years[38,47,48]. In some plant species with multicellular trichomes such as tomato and Artemisia, many HD-Zip and MYB transcription factors have been reported to regulate the trichome initiation[22,28,49–52]. In addition, trichome formation can also be influenced by many factors including hormonal signals, environmental cues, and aging programmes[38,53]. In *Arabidopsis*, AtTOE1 is involved in the age signalling and inhibits trichome initiation by repressing *GL1* expression. In tomato, *SlTOE1B* expression driven by *MTR1* promoter also caused a reduction of trichome number. Similarly, the number of trichomes was reduced by about 10% in *gcr1/2* double mutants. In contrast, trichome number was significantly increased in *sltoe1b* mutant and MTR1 promoter-driven GCR1/2 expression lines, suggesting a potential role for GCR1/2 and SlTOE1B in the initiation of multicellular trichomes in tomato. The increase in glands and specialised metabolites is accompanied by a decrease in the number of trichomes, which may represent a balance between defence and energy consumption, and may be a survival mechanism acquired by plants during evolution. Interestingly, the level of acyl sugar was dramatically

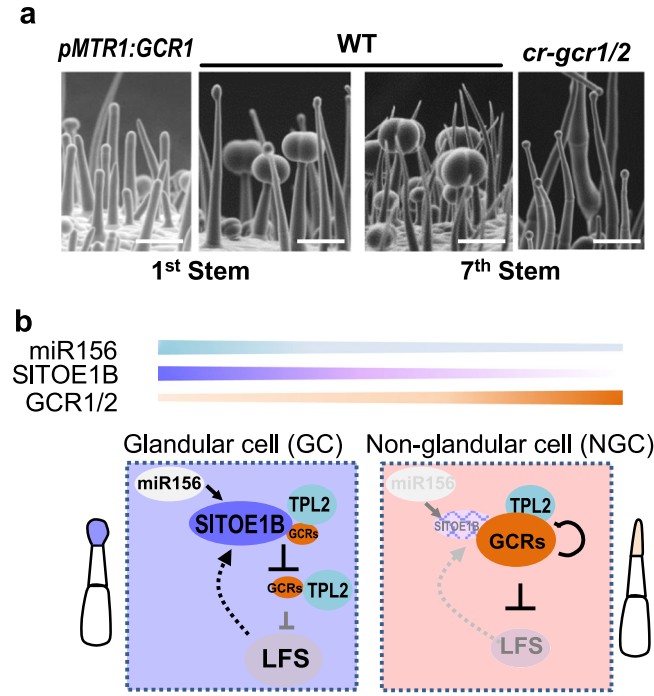

**a**

*pMTR1:GCR1*          WT          *cr-gcr1/2*

1st Stem          7th Stem

**b**

miR156
SlTOE1B
GCR1/2

Glandular cell (GC)          Non-glandular cell (NGC)

**Fig. 10 | Model explaining how *GCR1/2* spatiotemporally regulates the gland formation in plants. a** Trichome is an age-dependent feature in tomato. Trichomes appeared to be all glandular types including DGT and PGT at the juvenile stage (The second SEM image). In contrast, trichomes formed in adult plants include both NGT and PGT (The third image). However, forced expression of *GCR1* in trichome apical cells inhibited the gland formation (The first image). In contrast, when both *GCR1* and *2* were knocked out, supernumerary glands formed throughout the all developmental stages (The forth image). Bar: 50 μm. **b** The formation of glandular and non-glandular trichomes relies on the spatiotemporal expression of *GCR1/2*. Both genes were highly expressed in apical cells of NGT, but low expression in the apical cells of GT. This spatiotemporal expression of *GCR1/2* results from a combination of self-inhibition and *SlTOE1B* induced inhibition. Once the *GCR1/2* expression was significantly reduced, LFS was activated, which leads to the formation of gland heads. Both phase-related signals miR156 and SlTOE1B promotes glandular trichome formation.

increased (tens of fold) in *gcr1/2* double mutant and *pMTR1:SlTOE1B* lines (Supplementary Fig. 21). Therefore, GCR1/2 could be an important target for gene editing to increase acyl sugars for anti-insect resistance purposes.

## Methods

### Materials and plant growth conditions

We used Micro-Tom (MT), Alisa Craig (AC) and wild tomato accession *S. pimpine* LA1589 as wild type (WT) to generate the overexpression and CRISPR/Cas9 transgenic plants of GCR1 and GCR2. All wild type and transgenic plants were cultured in greenhouse with controlled temperature and light (16 h light at 26 °C, 8 h night at 22 °C). Tomato and tobacco plants used for protoplast isolation were grown on the 1/2 MS medium at 26 °C and 16 h light/8 h dark cycles. *N. benthamiana* used for BiFC assays were cultured in greenhouse with controlled temperature and light (16 h light at 26 °C, 8 h night at 22 °C).

### Vector construction and tomato transformation

For expression of GCR1/2 and SlTOE1B driven by MTR1 promoter, about 3 K promoter sequence of MTR1 fused with the full-length coding sequences of GCR1/2, rSlTOE1B and miR156B precursor and cloned into the pHellsgate8 vector (*pMTR1: GCR1-GFP, pMTR1: GCR2-GFP, pMTR1: rSlTOE1B-GFP, pMTR1: miR156B*). For knockout of GCR1/2 and TPL2 by CRISPR/Cas9, the target sites were designed using the online tool (http://crispr.hzau.edu.cn/CRISPR2/) and inserted into PTX

Vector (*cr-gcr1, cr-gcr2, cr-tpl2*). For expression pattern analysis of GCR1/2, about 3 K promoter sequence was fused with nuclear localisation signal (NLS) and Stadygold and inserted into the pHellsgate8 vector (*pGCR1: NLS-Staygold, pGCR2: NLS-Staygold*). For expression pattern analysis of SlTOE1B, about 3 K promoter sequence was fusion with NLS-Venus and then cloned into pHellsgate8 vector (*pSlTOE1B: NLS-Venus*). Primers used for vector construction were listed in Supplementary Table 12. The transgenic plants were constructed by *A. tumifaciens C58* mediated stable transformation[54]. PCR, qRT-PCR and Sanger sequencing were used to identify the positive transgenic plants. For lines with fluorescence proteins, we observe the presence of fluorescence in the plant trichomes through a fluorescence microscope (Leica TCS SP8X DLS).

### Phenotype analysis

Light microscopy images of stems, leaves and flowers were captured by the stereomicroscope (DFC550 LEICA). The trichome phenotype of wild type and transgenic plants was observed by scanning electron microscope (SEM, TM3030Plus, HITACHI, JAPAN). To reduce statistical errors, the WT and transgenic plants were grown under the same conditions, and images were captured in the same setting.

### Gus staining

To analyse the glandular cell identity of *gcr1/2* single and double mutants, we crossed those mutants with *pSlAT2: GUS-YFP* lines expressing in the glandular cell of DGT[55]. The leaves were stained in the GUS solution (10 mM EDTA disodium salt, 100 mM NaH$_2$PO$_4$, 0.5 mM K$_4$Fe(CN)$_6$, 0.5 mM K$_3$Fe(CN)$_6$, 0.1% Triton-X100, 0.5 mg/ml X-gluc (Golden Biotechnology, G1281C1, pH 7.0) at 37 °C for overnight. Then the stained tissues were immersed in destaining solution (80% ethyl alcohol and 20% acetic acid) for 30 min to remove chlorophyll and rehydrated with the gradient concentration of alcohol (60%, 40%, 20% and 0%) for 30 min in each gradient solution. All samples were observed under a differential interference microscopy (DIC, Nikon, ECLIPSE Ni-U).

### Expression analysis of GCR1 and GCR2

Shoot apical meristem of 9-day-old *pGCR1: NLS-Staygold* and *pGCR2: NLS-Staygold* transgenic plants were stained by propidium iodide (PIE) and imaged under a Leica TCS SP8X confocal laser scanning microscope (or a Carl Zeiss LSM880 confocal laser scanning microscope) with z-stack. The same setting and parameter was applied for all images. All images were processed by maximum intensity projection and measured the fluorescence intensity of Staygold using of Leica TCS SP8X DLS software or Zeiss LSM 880 software. No additional processing is done to the Staygold signal.

### Yeast-two-hybrid (Y2H)

The Matchmaker GAL4 Two-Hybrid System was used for screening a cDNA library and protein-protein interaction analysis in this study.

For screening the interacting protein of GCR1, we constructed a yeast cDNA library using tomato trichomes and shoot apical tissues of Micro-tom. The full-length coding sequence of GCR1 was cloned into pGBKT7 vector as a bait vector (BD-GCR1). Before screening the library, autonomous activation of BD-GCR1 was detected by culturing the yeast stain AH109 containing the BD-GCR1 and empty pGADT7 vector on the SD/-Ade/-His/-Leu/-Trp (QDO) agar medium. 1 μg BD-GCR1 vectors and 15 μl library vectors were co-transformed into yeast stain AH109 and cultured on SD/-Leu/-Trp (DDO) agar medium. After 3–4 days, the Co-transformed AH109 yeast cells were transferred to QDO agar medium and cultured for 3–4 days to screen positive colonies. Amplifying positive clones by PCR and performing Sanger sequencing. The sequences were then used to identify the corresponding interacting proteins in *S. lycopersicum* genome.

For protein–protein interacting, the coding sequences of examined protein pairs were inserted into pGBKT7 (bait vector) vector and

pGADT7 vector (prey vector) respectively. Pairs of recombinant constructs were Co-transformed into yeast stain AH109 and cultured on DDO medium for 3–4 days. The Co-transformed yeast cells were suspended in sterile ddH$_2$O to 1, 0.1 and 0.01 (OD$_{600}$), and then cultured on QDO medium for four days. For each combination and each concentration, three replicates were conducted, and one of the replicate was presented in the figure. All of used primers are listed in Supplementary Table 12.

## Quantitative RT-PCR (qRT-PCR) experiments

Trichomes was scraped and collected from stem after freezing in liquid nitrogen. Total RNA of trichomes, fresh leaves and stem was extracted with the Eastep® Super Total RNA Extraction Kit (Promega. ls1040). First-strand cDNA was synthesised from 1 μg of total RNA with the HiScript[1st] Strand cDNA Synthesis Kit (Vazyme Biotech co. Ltd/Cat#R111-01). Quantitative real-time PCR was performed with a CFX384 Real-Time system (BIO-RAD) by using ChamQ[TM] Universal SYBR qPCR Master Mix (Vazyme/Cat# Q711-02). PCR reaction system were as follows: 94 °C for 3 min; 40 cycles of 94 °C for 5 s and 60 °C for 30 s. PCR amplification specificity was verified by a dissociation curve (65 °C to 95 °C). SlActin 2 (Solyc11g005330) was used as reference gene to normalised gene expression. Primers are listed in Supplementary Table 12.

Quantitative RT-PCR of miR156B was detected by an end-point and real-time looped RT-PCR procedure[56]. First strand cDNA was synthesised from 1 μg of total RNA with miRNA[1st] Stand cDNA Synthesis Kit (by stem-loop) (Vazyme Biotech co. Ltd MR101-01/02). The U6 gene (Solyc12g056280) was used as reference gene to normalized miR156B expression.

## Protoplast isolation and luciferase (LUC) assay

The full-length coding sequences of examined proteins and mutant varieties were inserted into the pGreen II 62-SK vector (effector). The examined promoter sequences were cloned into the pGreen II 0800 vector (reporter). Leaves of tobacco and *cr-gcr1/2* were used to isolate protoplast by methods described previously[57]. Recombinant effectors and reporters were co-transformed into protoplasts at the ratio of 1:7 according to the method described previously[23]. After culturing overnight at room temperature, the LUC activity was detected using Dual-Luciferase®Reporter Assay System (Promega, Cat#E1910) and CYTATION5 image reader (BioTek). Three to five biological repeats were conducted for each combination. Error bars represent the SD. Statistical analysis is performed using *T*-test (and nonparametric tests) and significant differences were marked with Asterisks.

## Bimolecular fluorescence complementation (BiFC)

The full-length coding sequences of examined protein pairs were infused with C-terminal of yellow fluorescent protein (cYFP) or N-terminal of YFP (nYFP) respectively. Pairs of Agrobacterium strains GV3101 harbouring the recombinant plasmid and p19 were Co-infiltrated into abaxial side of leaves of 4-week-old *N. benthamiana* plants. We infiltrated one leaf per plant, and three individual plants were included for each combination. The leaves were checked by confocal microscopy 3–4 days post infiltration.

## Yeast-one-hybrid (Y1H)

The full-length coding sequences of examined proteins and mutant varieties were inserted into the pJG4-5 vector. The examined promoter sequences and motifs were cloned into the pLaczi vector. Pairs of recombinant constructs were Co-transformed into yeast stain EGY48 and cultured on SD/-Ura/-Trp agar medium for 3–4 days. The Co-transformed yeast cells were suspended in sterile ddH$_2$O to 5, 1 and 0.1 (OD$_{600}$), and then cultured on SD/-Ura/-Trp+X-gal (TransGen Biotech /GF201-01) medium for three days. Blue clones indicate that there is an interaction between the protein and the DNA sequence, while a white clone indicates no interaction. For each combination and each

concentration, three replicates were conducted, and one of the replicate was presented in figures.

## GST-pull down

The N-terminal of SlTOE1B and TPL2 were fused with GST (GST-SlTOE1B, GST-TPL2). The N-terminal of GCR1, GCR2, and SlTOE1B were fused with His (His-GCR1, His-GCR2, His-SlTOE1B). The recombinant vectors were transformed into the *Escherichia coli* strain BL21 (DE3) and the fusion proteins were expressed by inducing with 250 μm/L isopropyl-b-D-thiogalactoside (IPTG). The fusion proteins were purified using GST beads or His beads. The GST protein and GST fusion protein were incubated with GST MAGIC beads in incubating buffer (50 mM HEPES pH 7.5, 50 mM NaCl, 10 mM EDTA, 0.2% (v/v) Triton X-100, 10% (v/v) Glycerol, 2 mM DTT, 1 × protease inhibitor, 1 × PMSF) at 4 °C for 1.5 h. After washing two times with the washing buffer (50 mM HEPES pH 7.5, 150 mM NaCl, 10 mM EDTA, 0.1% (v/v) Triton X-100, 10% (v/v) Glycerol, 1 × PMSF), the samples were incubated with His fusion protein for 2.5 h. Finally, the samples were washed six times and boiled with SDS loading buffer for 8 min. The samples were detected by SDS-PAGE and immuno-blotted using anti-His antibody (1:3000 dilution, Abmart, Cat#M20001S) and anti-GST antibody (1:3000 dilution, Abmart, Cat#M20007S). All primers are listed in Supplementary Table 12.

## Biotin labelled DNA-IP Assays

DNA fragments were amplified by PCR using primers labelled with biotin. The streptavidin-agarose beads (Streptavidin Agarose, Cat#SA100-04) were washed three times using PBSI buffer (PBS buffer pH 7.4, 2 mM DTT, 1 × protease inhibitor, 1 × PMSF). Then incubated with 0.4 mg/mL DNA (Salmon sperm) and 1 mg/mL BSA for overnight at 4 °C. The samples were washed three times using PBSI buffer. Three μg labelled DNA fragments incubated with streptavidin-agarose beads for overnight at 4 °C. Finally, the samples were washed three times using PBSI buffer and incubated with 15 μg His-GCR1 and His-GCR2 fusion proteins, respectively, at 4 °C. After incubating for 4 h, the samples were washed six times and suspended in 40 μl 2 × Laemmli sample buffer (4% SDS, 20% Glycerol, 120 mM Tri-HCl pH 6.8) and then detected by SDS-PAGE and immuno-blotted using anti-His antibody (1:3000 dilution). His-GST protein was used as negative controls. All buffers used in this assay were found the previous paper[58].

## RNA-seq and transcriptome data analysis

Trichomes of *S. Pennellii* (LA0716) were scraped from the stems after freezing in liquid nitrogen. Stem epidermis removal trichomes was used as control. Three biological replicates for each line were performed. Standard RNA-sequencing libraries and sequencing were conducted by Novogene with the TruSeq Stranded Total RNA Library Prep Kit (Illumina) and an Illumina sequencing system on Hiseq 2500 platform (Illumina). The raw RNA sequencing reads was filtered to remove low-quality sequence, adaptors, duplications and calibrated low-quality bases of overlap range using fastp software. The clean reads were mapped onto the Slycopersicum genome (v3.0) using HISAT2 (v2.2.1) and assembled transcript by using stringtie (v2.1.7). The gene expression level was quantified by TPM (Transcripts Per Kilobase of exon-model per Million mapped reads). Differentially expressed genes (DEG) were identified by DESeq2 based on a threshold of log$_2$fold change > 1.

The transcriptome sequencing of trichomes of *pMTR1: GCR1-GFP* (Micro-Tom), *cr-gcr1/2* (Micro-Tom) and Micro-Tom wild-type was conducted as the above method.

Veen diagrams were conducted by Venny 2.1.0 (https://bioinfogp. cnb.csic.es/tools/venny/index.html). Heat-map of DEG was constructed by GraphPad Prism (V8.0).

## Acyl sugar measurement

Extraction and analysis of acyl sugars were conducted according to the previous method[59]. Two leaflets of juvenile stage (for *pMTR1: GCR1/2-*

GFP and *cr-lfs*) or adult stage (for *cr-gcr1/2* and *pMTR1: SlTOE1B-GFP*) were used to detect the acyl sugars content. Acyl sugars was extracted in 1 ml extractant (3:3:2 Acetonitrile: Isopropanol: water, 0.1% formic acid; 2 μg telmisartan (internal standard)) for 2 min. The supernatant was transfer to the LC-MS glass vial.

For analysing acyl sugars extraction, we used an LC-MS system, which is a Waters Acquity UPLC system coupled in tandem to a Waters photodiode array (PDA) detector and a SYNAPT G2-Si HDMS QTOF mass spectrometer (Waters, Manchester, UK). Gradient elution was achieved on a Waters Acquity UPLC HSS T3 column (100 × 2.1 mm, 1.8 μm) with water containing 10 mM ammonium formate PH2.8 (solvent A) and 100% acetonitrile (solvent B) at a flow rate of 0.3 mL/min. The column temperature was maintained at 40 °C. We adopted 21 min gradient elution procedure by reviewing the literature[60].

Components eluting between 1 and 14 min from the UPLC-QTOF MS system were processed in Progenesis QI (v2.1, Nonlinear Dynamics, Newcastle upon Tyne, UK) for data analysis. At least six types of acyl sugar were detected based on the previous NMR-elucidated structure of acyl sugar[61].

### Co-immunoprecipitation (Co-IP) assay
For CoIP assay, the full-length or fragment of target genes were fused with GFP or FLAG. Paired recombinant vectors were Co-transformed into tobacco protoplasts. After culturing overnight at room temperature, protoplasts of tobacco were collected in the protein extraction buffer (50 mM HEPES pH 7.5, 10 mM EDTA, 50 mM NaCl, 10% (v/v) Glycerol, 0.2% (v/v) Triton X-100, 5 mM DTT, 80 μM MG132, 1 × PMSF, and 1 × protease inhibitor cocktail). Protein extract was centrifuged at 13,000× g at 4 °C for 15 min and the super-natants were incubated with GFP-Nanoab-Magnetic (PGM050, LABLEAD Inc. Cat#GNM-50-2000) at 4 °C for 2–3 h. Then, the beads were washed three times with wash buffer (the same as the extraction buffer with 150 mM NaCl). The samples were resolved by SDS-PAGE and immuno-blotted using ProteinFind® Anti-GFP Mouse Monoclonal Antibody (1:1000 dilution, TransGen Biotech, Cat#HT801), ProteinFind® Anti-Flag Antibody (1:1000 dilution, PGM050, LABLEAD Inc. Cat#F1005), and anti-HRP Secondary antibodies (1:3000 dilution).

### CHIP-qPCR
ChIP assay was carried out as described previously[62]. Briefly, 2.5 g young leaves of 3-week-old *p35S:GCR1-GFP* transgenic plants were harvested and cross-linked in 1% (v/v) formaldehyde on ice in vacuum for 15 min. Nucleus was collected with Honda Buffer and then chromatin was fragmented by sonication on ice to an average DNA length of between 250–500 bp. Immunoprecipitation was done with anti-green fluorescent protein (anti-GFP antibody, Sigma, Cat#G6795) and protein G beads (Invitrogen, Cat#88848). Inputs and Chip samples were digested with Proteinase K at 65 °C overnight and immunoprecipitated DNA was purified by phenol extraction. The immunoprecipitated chromatin was analysed by qPCR. Data are presented as mean ± SD. All primers are listed in Supplementary Table 12.

### Phylogenetic analysis
We searched for protein sequences similar to GCR1 in Arabidopsis, tobacco and petunia in Phytozome (https://phytozome-next.jgi.doe.gov/) and Solyc04g049800 in TAIR (https://www.arabidopsis.org/). Phylogenetic trees were constructed by MEGA7 software. Sequence aligned by ClustalW and then phylogenetic analyses were conducted using the Neighbour–Joining (NJ) method.

### Accession Numbers
Sequence data from this article can be found in the Phytozome databases under the following accession numbers: *GCR1*, Sloyc02g076670; *GCR2*, Solyc03g006150; *NbGCR1* (*Nb2011*), Niben101SCF02336g02011;

*PeGCR*, Peaxi162Scf00940g00110; *SlTOE1B*, Solyc04g049800; miR156B, AT4G30972; *LFS*, Solyc05g013540; *TPL1*, Solyc03g117360; *TPL2*, Solyc08g076030; *TPL3*, Solyc01g100050; *TPL4*, Solyc03g116750; *TPL5*, Solyc07g008040; *TPL6*, Solyc08g029050; *ACTIN2*, Solyc11g005330.

### Statistics and reproducibility
All statistical analyses were conducted with GraphPad Prism 9.5 Software. No data were excluded from the analyses. *p*-values were obtained by unpaired two-sided *t*-test. Numbers of repetitions and replicates for each experiment were indicated in the legends.

### Reporting summary
Further information on research design is available in the Nature Portfolio Reporting Summary linked to this article.

## Data availability
The transcriptome data generated in this study have been deposited in NCBI under accession code PRJNA1074340 and PRJNA1074291. All data generated or analysed in this study including figures, Supplementary Figs. and supplementary tables are available. Source data are provided with this paper.

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

## Acknowledgements

The authors thank Dr. Zhenbiao Yang for critical comments. We thank Junqing Wu and Zhongquan Lin for technical assistance. This work was supported by the National Natural Science Foundation of China (32370376) to J. Chang and (32370354) to S. Wu, the Natural Science Foundation of Fujian Province (2023J01274) to J. Chang and Outstanding Youth Programme of Fujian Agriculture and Forestry University (Kxjq20004) to J. Chang.

## Author contributions

J. Chang and S. Wu. conceived and designed the experiments; J. Chang. S. R. Wu., and T. You performed most of the experiments and analysed the data; B. J. Sun. and J. F. Wang., performed vectors construction, protein purification and tomato stable transformation. B. J. Xu. performed vectors construction. X. C. Xu and Y. P. Zhang detected positively transgenic plants and managed materials.

## Competing interests

The authors declare no competing interests.
