## [Peer review file · Nature Communications]

Spatiotemporal formation of glands in plants is modulated by MYB-like transcription factorsREVIEWER COMMENTS

Reviewer #1 (Remarks to the Author):

In this manuscript, Chang et al. reported the molecular pathway that regulates the gland formation in the tip of the trichomes in tomato. The authors first identified the MYB transcription factors GCRs as the repressor of gland formation, while SITO1 is the activator of gland formation. They further analyzed the molecular network involving GCRs, SITO1, miRNAs, TPL, and LFS. This complex gene/protein network ensures the developmental regulation of gland or non-gland trichome formation. Overall, the data are carefully presented. The story will improve our understanding of how cell differentiation is regulated using the tomato trichome/gland as the model. I have several suggestions for the authors to improve the manuscript.

1) The manuscript will benefit from the modification/improvement of Abstract and Introduction. The Abstract and Introduction parts are relatively short and the information (for examples, the current knowledge of trichome/gland development) is limited. In addition, the gene/protein interactions or regulations might be clearly stated in Abstract.

2) Figure 1E, the GUS staining is not clear. Are there any other pictures showing the GUS staining result? Or the authors may make some annotations on the GUS staining result.

3) The models in Figure 3 and Figure 5 might be combined. Or alternatively, the model might be listed as a Figure at the end of the story.

4) Some data in the Supplemental Figures are interesting and might be moved to the main text. For examples, some phenotype data in Figure S3, miR156 data in Figure S8, and the phenotype of the *gcr1/gcr2/lfs* triple mutant in Figure S9.

5) Figure 1H. Spatio-temporal expression of GCR1. This is an important result of the story. Are there more pictures showing the pattern (which might be listed in Supplemental data)?

Reviewer #2 (Remarks to the Author):

The manuscript explores the activity of two tomato genes (GCR1/2) and an interacting partner (SITOE1) with respect to their role in the formation of glandular trichomes. While potentially of interest, the manuscript fails to put their findings in the context of what is already known about this process in tomato (their plant of study) and Arabidopsis (which can be forced to form multi-cellular trichomes).

The manuscript suffers from some major deficiencies, not just by failing to integrate results with existing knowledge, but also by providing figures in a format that makes it impossible to evaluate. Often general claims are made with only one image being shown and no quantitation. Thus, in multiple instances I had to “believe” what the authors said to be able to continue with the manuscript.

Comments

1. The summary and introduction are to some extent misleading – the GCR1/2 genes don't appear to determine which cells will become a trichome, but rather convert unicellular trichomes into multi-cellular trichomes. This is by no means clear in the abstract. This distinction is very important, because a gene (SIAMESE) that controls this conversion was already reported in Arabidopsis almost two decades ago (<https://www.ncbi.nlm.nih.gov/pmc/articles/PMC1693949/>). It is disappointing that this study was not cited in this paper.
2. What is the pMTR1 promoter? Not clear why it was used or what its expression pattern is.
3. Quantitation of the number of total, glandular, and non-glandular trichomes in the experiments shown in Fig. 1I and Fig. S3F must be shown
4. Figs. 2A, 4D and 4E are unacceptable – at the very least cell dilutions in the same plate for each of the interactions needs to be presented. It is impossible from the figure or legend to understand which protein is fused with each domain (DNA-binding or activation).
5. Fig. 2B is lacking additional controls and a quantitation of the results as a single example is shown.
6. Entire gels with molecular weight markers need to be shown for Fig. 3C. The figure is too edited to be able to evaluate. Also, the figure needs to be clearer with respect of what was used for pull-down and for the western.
7. Multiple images and quantitation of the expression of pSITOE1:GUS is needed – in the image shown, there appears to be GUS expression not just in the gland, but also in stalk of the gland at lower level.
8. The authors refer to Fig. 2E to mention that both digitate trichomes (DT) and peltate trichomes (PT) increased – however, this is neither indicated nor quantified, and what these types of trichomes was never explained in the manuscript.
9. The manuscript requires very major English language editing as often the grammar made the sentences have equivocal meaning.

10. Unclear how the RNA expression experiments were done in Fig. 3A – error bars are too small for biological replicates, particularly after error progression to obtain relative fold levels. Why is the length of the trichomes so much larger in #2? This was never explained.
11. The Y1H and biotin-I experiments (Figs 3C & D) are insufficient to demonstrate binding of GCR1/2 to their own promoters. Chromatin immunoprecipitation experiments are necessary. Same applies to Fig. 4.
12. It is my understanding that GCR1/2 are part of a large gene family (this was never mentioned or explained) – the manuscript is misleading in saying that JASPAR predicted the binding of GCR1/2 to the motifs presented. What JASPER does is to say that at least one member of this large family of regulatory proteins has been shown in at least one species to bind a motif with that sequence. This also applies to the analyses of the FLS promoter.
13. Because of concerns on how the results are presented, it is difficult to evaluate whether LFS is indeed directly regulated by GCR1/2 or not. The authors never explained what LFS is and how understanding its regulation by GCR1/2 helps understand how GCR1/2 contribute to the conversion of single-cell to glandular trichomes.
14. The discussion in the context of AtGL1 is confusing and misleading – AtGL1 is involved in selecting which protodermal cells will become a trichome, a completely different phenomena from the one discussed here – in fact, the authors at times seem confuse trichome initiation from glandular trichome establishment.
15. What is the relationship of GCR1/2, SITO1 and Woolly? Given how much is already known about Woolly, it is unfortunate that it was mentioned in the intro but none of the experiments conducted were aimed at finding a relationship. This is a very significant shortcoming of this study.

Reviewer #3 (Remarks to the Author):

The manuscript titled with “Spatiotemporal formation of glands in plants is modulated by MYB-like transcription factors” by Chang et al. demonstrates GCR MYB transcription factors interacted with and inhibited the gland-promoting regulator AP2/ERF transcriptional factor SITO1, and both of them interacted with transcriptional co-repressor SITPL2. By analyzing the published transcriptome data, the authors selected two uncharacterized MYB transcription factors, GCR1 and GCR2, which were predominantly expressed in the glandular trichomes. The *gcr1* single mutants exhibited the conversion of some non-glandular trichomes to glandular trichomes, and the double mutants *gcr1 gcr2* generated supernumerary glands, and frequently produced peltate trichomes with supernumerary glandular heads. GCR1 was expressed in tomato trichomes. The ectopic expression of GCR1/2 entirely inhibited gland formation, which was opposite to the phenotype of *gcr1 gcr2* mutant. The authors performed yeast two-hybrid screen for GCR1 interactors in using a cDNA library, and found that SITO1 interacted with GCR1. The interaction was confirmed by yeast two-hybrid, BiFC and pull-down assays. The overexpression of

miR172-resistant-rSITOE1 caused the increased glandular cells in digitate trichomes and peltate trichomes. The authors showed that SITOE1 formed complex with GCR1/2 and SITPL2 to directly repress GCR1/2. They further showed that GCR1/2 interacted with SITPL2 and directly repressed the expression of LFS is involved in the control of trichome formation.

The manuscript provides interesting results that showed two MYB transcription factors GCR1/2 redundantly promote glandular cell fate during tomato trichome formation. In this process, GCR1/2 act as repressors to inhibit LFS and itself by forming complexes with SITOE1 and SITPL2. However, I think that the writing and data presentation have a lot of errors. And more data and experiments are required to support the working model and the conclusion of the manuscript. The detailed concerns were listed below.

Major concerns:

1. The introduction is so simple that many important information facilitating the understanding of the results are lack. In the introduction, the main information that most readers may not know should be provided. For examples, how many types of trichomes in the tomato? What is the main related progress in the molecular mechanisms underlying the formation of different trichomes? What is the EAR motif? What are the known functions of SITOE1 and SITPL2, and etc.
2. At least two alleles of gene mutants should be analyzed or the complementation should be provided to support that the loss of the gene function led to the defects of trichome formation.
3. The authors described that “trichomes was replaced by ever-growing trichomes with supernumerary glands” “the short glandular trichomes (peltate trichomes, PT) often produced supernumerary glandular heads in double mutants”. How many? How often? I suggested that different kinds of trichomes but not only non-glandular trichomes and glandular trichomes should be statistically analyzed to resolve the question.
4. What is the trichome phenotypes when the function of SITOE1 is compromised. The ectopic expression of SITOE1 could not validate the real function of SITOE1 in the control of trichome formation.
5. The binding of GCR1 and GCR2 in the promoters of GCR1 and GCR2 should be tested using ChIP analysis in vivo.
6. Does SITOE1 bind to the promoters of GCR1 and GCR2?
7. Genetic interaction between TOE and GCR1 is necessary, what is the phenotype of double mutant *gcr1 sltoe1*?
8. GCR1 and GCR2 could directly interact with SITPL2 via its own EAR motifs. Why do GCR1 and GCR2 need to recruit SITPL2 by interacting with SITOE1? Which domain of GCR1 and GCR2 is responsible for their interaction with SITOE1? And which domain of SITOE1 is responsible for its interaction with GCR1 and GCR2?
9. “Furthermore, we found that the expression of GCR1/2 was strongly inhibited in *pMTR1: rSITOE1-GFP*, but interestingly became significantly enhanced in *cr-gcr1/2*”. Does the repression of GCR1/2 by SITOE1 require GCR1/2? How about the expression level of GCR1/2 in the loss of function mutant *sltoe1*?

10. The binding of GCR1 and GCR2 in the promoter of LFS should be also proved using CHIP analysis in vivo.

11. BiFC, LCI, or Co-IP should be used to verify that the protein interactions in vivo (like GCR1 and GCR2 interacting with TPL2, and SITO1 interacting with TPL2).

12. Fig.3B and Fig.5C, the control combinations including GCR1/2+rSITO1mEAR and GCR1mEAR+TPL2 should be included.

Minor concerns:

1. Line 49-51, No methods or description were found in “Materials and methods” in screening genes with the high expression level in the glandular trichomes. The authors could describe here and cited the published reference of RNA-seq data here.

2. Line 58, GCR1/2 have three EAR motifs as shown in Fig. S1D, so here “motif ” should be “motifs”.

3. The different mutants of *gcr1/2* and the double mutants should be described in the text. What are the differences of phenotypes between them?

4. Line 66, “trichomes in double mutants became significantly elongated”. The length of the trichomes and the number of cells consisted of trichomes should be analyzed and statistically calculated to conclude that the difference is significant.

5. Figure 1E, the control should be provided. That is, the image of glandular cell marker line in wild type and single mutants should be added. In the Fig. legend, “glandular cell maker line” has a typo. Maker should be marker?

6. Figure 1H, “with the expression stronger in apical cells of the non-glandular trichomes than glandular cells of glandular trichomes”. Which image in Fig. 1H is non-glandular trichome or glandular trichome? From the image, I found that in the middle image, the fluorescence of apical cells was also strong. How about the expression of GCR2? Here, the author did not give the data of time course, so in the figure legend, “Spatio-temporal expression of GCR1” is not accurate.

7. Figure 1I, the trichome phenotypes on the hypocotyl should be also statistically analyzed. Since in Fig.1A appear to present the trichome defects in the leaves (the authors should be provided the organ information in text and figure legend), the phenotype of trichomes in leaves of pMTR1:GCR1-GFP should be also statistically analyzed.

8. Line 90-91, “yeast two-hybrid experiments showed that GCR1/2 mainly interacted with SITO1 through the C-terminus”, in the Fig. 2. Legend “Y2H assay. GCR1/2 interacted with SITO1 through its N-terminal domain.” From the Fig 2A, the interaction domain should be C-terminus.

9. I suggested that the authors indicate the different kinds of trichomes in the phenotypic images of Figures.

10. Line 93, “Spatiotemporal expression analysis of pSITO1: GUS revealed that SITO1 was highly expressed in the glandular cells of glandular trichomes.” No time course GUS staining were shown, “Spatiotemporal expression” is not accurate.

11. Line 94 "SITOE1" and many gene names in the manuscript are not italic. Gene names should be italic.
12. Many abbreviations have no full names. The abbreviations used at the first time should be given full names in the Abstract and the main text of the manuscript.
13. Figure 2E, the phenotype in the image should be statistically analyzed.
14. Line 129-130, "SITOE1 acts as the transcriptional repressor via the interaction between its bearing EAR motif and TPL in Arabidopsis". Here, SITOE1 should be TOE1 in Arabidopsis?
15. The Figure legends for Fig.3B and Fig.5D are missed.
16. In Fig.4G, the phenotype of *gcr1/2* and *lfs* should be included as the controls.
17. The Discussion is not sufficient.
18. Fig 1A-D lacks the scale bar for the enlarged images, and Fig 1F lacks significance analysis.
19. Fig 2E lacks significance analysis. How many biological replicates were performed?
20. Fig 2D legend misses the unit after the bar. (line 590, "Bar: 25.")
21. How many transgenic lines were obtained for GUS analysis in Fig 2D?
22. The three replicates of WT should be put together in Fig S3B, Fig S9, and the significant analysis should be added.
23. The significant analysis should be performed in Fig 3A.
24. "Figure S9. Genotype and phenotype of *gcr1/gcr2/lfs* triple mutant". No data about the triple mutants was found in the Fig. S9.

Reviewer #4 (Remarks to the Author):

This manuscript investigates a transcription factor regulatory network in tomato that is involved in the differentiation of glands from capitate trichomes (type IV). Two MYB TFs (*GCR1/2*) were found as repressors of the gland cell differentiation. They are antagonized by another TF (*SITOE1*) which represses their expression and they repress *LFS*, which is a promoter of gland formation.

The data presented is of high interest for people working on tomato trichomes, and provide insight as to why cultivated tomato has few capitate trichomes in adult leaves. However, the quality of the execution and presentation of the data as well as the quality of the writing need much improvement. This makes for difficult reading. Figure legends are often incomplete or imprecise.

Major Comments:

- 1) The high expression of *GCR1* and *GCR2* in wild tomatoes (*S. pennellii*) contradicts the role of *GCR1* and *GCR2* as repressors of type IV trichomes, because *S. pennellii* has extremely high density of type IV

trichomes, as the authors rightly point out. This is certainly concerning and I am wondering if the authors have an explanation for this.

2) What is the pMTR1 promoter? I could not find any information on it in the text.

3) Line 158-162: the interpretation is not consistent with the genotypes of the lfs mutants. According to Fig. S9, both mutants are bi-allelic with mutations that result in the same aberrant proteins (in each mutant). Therefore no phenotypic difference would be expected between those two mutants. This brings another question: are the plants characterized the first generation transformants or the next generation? If they are from the next generation, then segregation can occur, but no simple heterozygote will be produced, only homozygous for each individual mutation or double heterozygote (like the parent).

4) Since type IV trichomes produce acyl sugars, it would be essential to measure these in all mutants characterized here.

5) An accession number for the transcriptome data produced in this manuscript must be provided.

6) There is a number of issues with the figures and presentation of the data:

a. Figure 1: as there are several types of glandular trichomes (type VI, type I/IV and type VII, the latter could be ignored). A much more specific counting of the different types is required. The authors should adopt the accepted nomenclature for the different types of trichomes (type I until VII).

b. Fig. 1: in panel H the bright field image should be provided. GCR1 appears to be expressed not just in the tip cell but in the stalk cells of the trichomes as well and in type VI glandular trichomes. Which raises the question, why when GCR1 is overexpressed it should affect type VI trichomes in cotyledons as shown in panel I?

c. Figure 2 (and others): Please provide complete western blots and coomassie gels with visible size markers for all IP and pull down experiments.

d. Fig. 2: provide trichome counts for panel E

e. Fig. 3: was the pull down done with purified GCR1? A coomassie gel should be shown. If not purified, a control with extract from a strain with the empty vector should be made.

f. Fig. 3: there is a problem with the legend. Panel B is missing, legends of panels have been mixed up.

g. Figure 4: again provide trichome counts. And same remark as in Figure 3 for the DNA IP assays.

h. Figure 5: Panel E is not explained. Where are the lfs mutants? What is cr-gcrs? There is something missing in the legend.

i. Fig. S2: The mutations in GCR genes in the different tomato backgrounds (panel D) must be provided.

j. Fig. S3: In panel A pMTZR1 confers expression in the whole trichome. Why is it only in the tip cell when fused to GCR1 (panel C). Again, provide trichome counts.

k. Please provide source organisms for NbGCR and PeGCR1.

l. Fig. S3. The lines characterized here are not the same as in Fig. 3. when they should be.

m. Fig. S7: provide full western blots with size markers visible and coomassie with size markers clearly labelled.

n. Fig. S8: trichome counts are need. Was the expression determined from whole leaves or just trichomes (panel E). Panel F: the value of the scale bar cannot be right.

o. Fig. S9: see comment 2) on the interpretation of the *lfs* mutants. And the legend says “phenotype of *gcr1/gcr2/lfs* triple mutant”. I only see *lfs* single mutants.

p. Fig. S9: Microtom has hardly any type IV trichomes on adult leaves. Are the pictures form cotyledons?

Other comments:

1) is miR172 present in tomato? Corresponding references should be cited.

2) Line 200: why should the repression via TPL2 be epigenetic? I did not see any evidence for this in the manuscript.

REVIEWER COMMENTS

Reviewer #1 (Remarks to the Author):

In this manuscript, Chang et al. reported the molecular pathway that regulates the gland formation in the tip of the trichomes in tomato. The authors first identified the MYB transcription factors GCRs as the repressor of gland formation, while SITO1 is the activator of gland formation. They further analyzed the molecular network involving GCRs, SITO1, miRNAs, TPL, and LFS. This complex gene/protein network ensures the developmental regulation of gland or non-gland trichome formation. Overall, the data are carefully presented. The story will improve our understanding of how cell differentiation is regulated using the tomato trichome/gland as the model. I have several suggestions for the authors to improve the manuscript.

1) The manuscript will benefit from the modification/improvement of Abstract and Introduction. The Abstract and Introduction parts are relatively short and the information (for examples, the current knowledge of trichome/gland development) is limited. In addition, the gene/protein interactions or regulations might be clearly stated in Abstract.

We thank the reviewer for the advice. We have expanded the introduction part as suggested.

2) Figure 1E, the GUS staining is not clear. Are there any other pictures showing the GUS staining result? Or the authors may make some annotations on the GUS staining result.

We have replaced the previous images. In the new figure, we added the control (the staining of DGT (digital glandular trichomes, type I and VI) and NGT (non-glandular trichomes, type II, III and V in WT) and marked trichome type in figures.

3) The models in Figure 3 and Figure 5 might be combined. Or alternatively, the model might be listed as a Figure at the end of the story.

Thanks for your suggestions! We removed the model in Figure 3 and kept the model in Figure 5 (now in figure 7).

4) Some data in the Supplemental Figures are interesting and might be moved to the main text. For examples, some phenotype data in Figure S3, miR156 data in Figure S8, and the phenotype of the *gcr1/gcr2/lfs* triple mutant in Figure S9.

Thanks for your suggestions! We have moved the images and quantification data of GCR2, PeGCR and NbGCR to the Figure 1 and added the phenotype and quantification data of WT, *gcr1/2* double mutant, *lfs* mutant and *gcr1/2/lfs* triple mutant in Figure 5.

5) Figure 1H. Spatio-temporal expression of GCR1. This is an important result of the story. Are there more pictures showing the pattern (which might be listed in Supplemental data)?

We have added images and quantification data of another lines to Figure 1H and moved the previous images and quantification data to Supplementary Figure 5.

Reviewer #2 (Remarks to the Author):

The manuscript explores the activity of two tomato genes (*GCR1/2*) and an interacting partner (*SITOE1*) with respect to their role in the formation of glandular trichomes. While potentially of interest, the manuscript fails to put their findings in the context of what is already known about this process in tomato (their plant of study) and *Arabidopsis* (which can be forced to form multi-cellular trichomes).

Response: Thank you for pointing this out! The manuscript was initially submitted to *Nature* and was later transferred to *Nature Comm*. So the last version was in a concise format, and we now re-wrote the manuscript, with a particular focus on integrating the findings to the bigger context of the field according to the comment.

The manuscript suffers from some major deficiencies, not just by failing to integrate results with existing knowledge, but also by providing figures in a format that makes it impossible to evaluate. Often general claims are made with only one image being shown and no quantitation. Thus, in multiple instances I had to “believe” what the authors said to be able to continue with the manuscript.

Response: Thanks for raising this issue. We have strengthened our phenotype analysis by adding additional biological duplications and quantification data, and validated protein-protein, protein-DNA interactions with other systems. For example, we performed quantification of trichome of all materials in the revised version (Fig 1E and K, Fig. 3E and G, Fig. 4B, Fig. 5I, Fig. S15, Fig. S15). We provided more images of independent biological replicates of p*GCR1*: NLS-Stadygold (Fig. 1H and Fig. S5). We further provided p*SITOE1*: NLS-Venus plants for *SITOE1* expression pattern analysis and quantified the expression level of *SITOE1* in the trichomes (Fig. 3A). We re-validated the interactions (*GCR1/2* and *SITOE1*; *GCR1/2* and *TPL2*; *SITOE1* and *TPL2*) using both BiFC and CoIP (Fig. 2, Fig. 6, Fig. S9, Fig. S14, Fig. S17) and employed ChIP-qPCR to verify the binding of *GCR1/2* to the promoters of their own and *LFS* (Fig. 4F and Fig. 5F). Lastly, we added more labels to the images to make them easier to understand, e.g., we explained and marked the different trichomes in Figure 1 and legend; we also labelled the fusion with DNA-binding domain (BD) and DNA-activation domain (AD) in the Yeast hybrid experiments; we marked the marker size in the pull down, CoIP and Biotin-IP assay and put in more detailed information in the figure legend to help the understanding of the results. We hope these improvements can help the reviewer to evaluate the results.

Comments

1. The summary and introduction are to some extent misleading – the *GCR1/2* genes don't appear to determine which cells will become a trichome, but rather convert unicellular trichomes into multi-cellular trichomes. This is by no means clear in the abstract. This distinction is very important, because a gene (*SIAMESE*) that controls this conversion was already reported in *Arabidopsis* almost two decades ago (<https://www.ncbi.nlm.nih.gov/pmc/articles/PMC1693949/>). It is disappointing that this study was not cited in this paper.

We appreciate the insightful comment from the reviewer, and we are sorry for not citing the mentioned paper. Different from the uni-cellular trichome in *Arabidopsis*, tomato trichomes undergo multiple rounds of cell division and the apical cells differentiate into different types of cells (including glandular cells) once the division stops. In *gcr1/2* double mutants, over 96% of trichomes became glandular trichomes.

In addition, the division seemed to be unceasing, often resulting in 30 cells in a trichome cell file. In contrast, there is about 8 cells in long digital cells (type I) in WT. Therefore, we conclude GCR1/2 presumably have two functions: to prevent the formation of glandular cells; and to prevent the excessive cell division in apical cells. In this study, we mainly focus on dissecting GCR1/2 function in the formation of glandular cells. As suggested by the reviewer, we added the discussion about the potential role of GCR1/2 in cell division and cited the related literature. We also re-wrote the abstract and introduction to make our conclusions clear.

2. What is the pMTR1 promoter? Not clear why it was used or what its expression pattern is.

Sorry for not making this clear in the last submission. Promoter of MTR1 was previously reported by our group (Wu, Dev Cell 2023), which is specifically active in tomato trichomes. In particular, pMTR1 activity is higher in apical cells of tomato trichomes, which makes it an ideal promoter to drive the specific expression of a gene in apical trichome cells. In the revised manuscript, we added the expression pattern of pMTR1 in SFig. 6A.

3. Quantitation of the number of total, glandular, and non-glandular trichomes in the experiments shown in Fig. 1I and Fig. S3F must be shown.

We have performed the quantification of the trichomes shown in Fig. 1I and Fig. S3F. Combined the opinions from reviewer 3 and 4, we quantified all seven type trichomes. To quantitatively evaluate the glandular trichome phenotype, we divided all trichomes into three categories: NGT (non-glandular trichomes including type II, III and V), DGT (digital glandular trichomes including type I and IV) and PGT (peltate glandular trichomes including type VI and VII) (Fig. S1A). Also based on the comment from reviewer 1, we combined the Fig. 1I and Fig. S3F and all the results are now in Fig. 1J-K in the revised paper.

4. Figs. 2A, 4D and 4E are unacceptable – at the very least cell dilutions in the same plate for each of the interactions needs to be presented. It is impossible from the figure or legend to understand which protein is fused with each domain (DNA-binding or activation).

Thanks for the suggestion. Based on the comment, we re-performed the interaction tests with gradient cell dilutions for all yeast hybrid assays shown in the manuscript (now in Figs. 4D, 5D, 5E, S8C, S9A, S14C and S17B). The dilutions were set as 1, 0.1, and 0.01 (OD₆₀₀). For each combination and each concentration, we included three replicates, and one of the replicate was presented in the figure. We also added detailed information for all yeast hybrid assays in the figures and figure legends.

5. Fig. 2B is lacking additional controls and a quantitation of the results as a single example is shown.

Thanks for the suggestion. We added the controls accordingly in the revised manuscript. We presented two negative control combinations including: target protein fused with nYFP + empty vector with cYFP; target protein fused with cYFP + empty vector with nYFP.

As there are a large number of combinations are tested, we therefore replaced the protoplast system with the tobacco leave infiltration. Both techniques currently are widely used. We infiltrated one leaf of each plant, and 3 individual plants were

included for each combination. We have added the detailed information of how the experiments were conducted in the materials and methods section in the revised manuscript.

6. Entire gels with molecular weight markers need to be shown for Fig. 3C. The figure is too edited to be able to evaluate. Also, the figure needs to be clearer with respect of what was used for pull-down and for the western.

We re-cropped the original complete gel images, and marked the molecular weight of the markers (now in Fig. S8D). We also provided the unprocessed images at the end of this file. What Fig. S8D presents is the GST-pull down assay, in which GST beads were used to co-immunoprecipitation and His antibody was used to detect the sample. The information has also been added properly in the revised manuscript.

7. Multiple images and quantitation of the expression of pSITOE1:GUS is needed – in the image shown, there appears to be GUS expression not just in the gland, but also in stalk of the gland at lower level.

Thanks for the comment! To quantify the expression level of SITOE1, we constructed another reporter line of pSITOE1: NLS-Venus. We examined over 10 independent transformants, and found SITOE1 is mostly expressed in the glandular cells. Occasionally weak expression of SITOE1 can also be observed in the cell just beneath the gland of DGT and PGT. However, we never detected the SITOE1 expression in non-glandular trichomes, nor in the more basal stalk cells of glandular trichomes. These observations support the conclusion that SITOE1 expression is correlated with the gland formation.

8. The authors refer to Fig. 2E to mention that both digitate trichomes (DT) and peltate trichomes (PT) increased – however, this is neither indicated nor quantified, and what these types of trichomes was never explained in the manuscript.

Thanks for the comment. We quantified the trichomes in pMTR1:SITOE1 lines. The results showed that when SITOE1 expression increased by 2 folds, all non-glandular trichomes (types II, III and V) disappeared and were replaced by digital glandular trichomes (type I and IV). When SITOE1 expression was increased by 10 folds, glandular cells of both DGT and PGT increased. We have added more glandular trichomes images in Fig. S10. We added explanation for different types of trichomes in the supplementary figure 1A.

9. The manuscript requires very major English language editing as often the grammar made the sentences have equivocal meaning.

Thank you for the comment. We have thoroughly revised the manuscript and have it proof-read by professional language service.

10. Unclear how the RNA expression experiments were done in Fig. 3A – error bars are too small for biological replicates, particularly after error progression to obtain relative fold levels. Why is the length of the trichomes so much larger in #2? This was never explained.

Sorry that we did not explain the figure clearly in the last submission. In Fig. 3A, the X-axis represents different individual plants (biological replicates). The error bars of the values along the Y-axis are actually technical replicates, and that is why the error bars appear to be small.

For the RNA expression analysis in Fig. 3A, we first crossed pMTR1: rSITOE1-GFP and pMTR1: GCR1-GFP lines, which resulted in 17 independent F1 plants. We extracted the RNA of young leaves of each individual F1 plants for qRT-PCR.

The long trichomes in #2 line is type I trichome, which is one of the largest trichome with multicellular base among the seven types (I-VII). In the revised manuscript, we re-performed all observations and conducted trichome quantifications of individual plants. The results are now shown in Fig. 4A. Among these lines, we found the varied levels of SITOE1-GFP and GCR1-GFP are closely associated with the gland formation. Compared with the #13 line (previous #6 line) in which both SITOE1-GFP and GCR1-GFP have relatively low expression, #2 and #6 (previous #2 and #3) have higher SITOE1-GFP expression and more glandular trichomes. #4 and #5 (previous #10 and #17) have higher SITOE1-GFP and GCR1-GFP expression, but formation of DGT and PGT was significantly inhibited. This result show that GCR1 is epistatic to SITOE1 in the gland formation pathway.

11. The Y1H and biotin-I experiments (Figs 3C & D) are insufficient to demonstrate binding of GCR1/2 to their own promoters. Chromatin immunoprecipitation experiments are necessary. Same applies to Fig. 4.

To address this comment, we performed ChIP-qPCR using 35S:GCR1-GFP transgenic plants to test whether GCR1 binds to the promoter of its own and LFS. The results support the conclusion drawn from Y1H and biotin-I experiments, and are now shown in Figs 4F and Figs 5F in the revised manuscript.

12. It is my understanding that GCR1/2 are part of a large gene family (this was never mentioned or explained) – the manuscript is misleading in saying that JASPAR predicted the binding of GCR1/2 to the motifs presented. What JASPER does is to say that at least one member of this large family of regulatory proteins has been shown in at least one species to bind a motif with that sequence. This also applies to the analyses of the FLS promoter.

Thanks for the comment. The homologues genes of GCR1/2 in Arabidopsis are AT2G38300 and AT2G40260, and all of them belong to GARP family (Safi, A., et al., 2017, *Curr Opin Plant Biol*). In JASPER database, the potentially binding site of AT2G38300 and AT2G40260 is HHHATTCYHHH and HHWHATTCYHHH with core ATTC sequence. We therefore speculate that GCR1/2 can also bind to the similar sites.

We then searched for such sequences in the promoter regions of GCR genes, and found 13 such sequences in the 3kb upstream of GCR1 and 10 in the 3kb upstream of GCR2. So we first conducted Y1H screen and found GCR1/2 can bind to most of the predicted motif. We next employed Y1H, Biotin-IP and ChIP-qPCR to test such interaction in vitro and in vivo. Similarly, we identified 12 potential motifs in the promoter of LFS and we verified GCR1 can indeed bind to LFS promoter.

To avoid the misleading description, we revised the corresponding part and changed the description to: based on the binding site for GCR homologues genes in Arabidopsis (AT2G38300 and AT2G40260) provided by JASPER database, we speculate that GCR1/2 may potentially bind to cis-element containing HHHATTCYHHH or HHWHATTCYHHH.

13. Because of concerns on how the results are presented, it is difficult to evaluate whether LFS is indeed directly regulated by GCR1/2 or not. The authors never

explained what LFS is and how understanding its regulation by GCR1/2 helps understand how GCR1/2 contribute to the conversion of single-cell to glandular trichomes.

Sorry for not explaining LFS well. This gene encodes an AP2 transcription factor, which has been reported to promote the fate of tomato peltate trichomes (Wu et al., *Developmental Cell*, 2023). GCR1/2 are highly expressed in non-glandular cells and have extremely low expression in glandular cells. It is possible that highly expressed GCR1/2 in non-glandular cells suppress the positive regulator of gland formation. Our results support that GCR1/2 can recruit TPL2 to LFS promoter to repress its expression. In the apical trichome cells about to become glandular cells, absence of GCR1/2 activity allows for the expression of LFS that eventually promotes the gland formation. We have revised the manuscript to clarify these regulations.

14. The discussion in the context of AtGL1 is confusing and misleading – AtGL1 is involved in selecting which protodermal cells will become a trichome, a completely different phenomena from the one discussed here – in fact, the authors at times seem confuse trichome initiation from glandular trichome establishment.

Sorry for the confusion. As AtGL1 is also a MYB gene and it has been shown that its expression can be repressed by TPL-HDA module, we speculate that a similar mechanism may exist for GCR1/2 inhibition by SITO1. In fact, the mechanism reported to regulate uni-cellular trichome formation in *Arabidopsis* appears to be quite different from the one controlling multi-cellular trichomes. It has been shown that AtGL1, the key regulator of Arabidopsis trichome, is not involved in trichome development in multicellular trichomes of tobacco (Payne et al., 1999). So here we try to discuss the similarity of the regulation at molecular level. To avoid confusion, we revised the discussion accordingly.

Payne, T., Clement, J., Arnold, D. & Lloyd, A. Heterologous myb genes distinct from GL1 enhance trichome production when overexpressed in *Nicotiana tabacum*. *Development* 126(4), 671–682 (1999).

15. What is the relationship of GCR1/2, SITO1 and Woolly? Given how much is already known about Woolly, it is unfortunate that it was mentioned in the intro but none of the experiments conducted were aimed at finding a relationship. This is a very significant shortcoming of this study.

Thanks for bringing this issue. We agree that it is interesting to know how *Woolly* can affect the *GCR* mediated regulation. As *Woolly* is the master regulator that functions in the upstream of regulatory cascade of trichome development, it is likely *GCR1/2* are downstream factors. However, their functions seem to be different at the spatiotemporal level. We only observed a marginal reduction (10%) of trichome density in *gcr1/2* double mutants, suggesting *GCR1/2* maybe not play the significant role in trichome initiation as *Woolly* does.

To further address this comment, we performed the following analyses:

1. In transcriptome, we found the upregulation of *GCR1/2* in *Woolly* dominant mutant, whereas *GCR1* expression became reduced in *cr-wo* mutant. However, we think this is probably due to the dramatic change of trichome numbers in different backgrounds.

2. We next crossed *Woolly* dominant mutant with *gcr1* mutant, and found *GCR1* mutation does not affect trichome numbers in *Woolly* dominant mutant. Instead, many apical cells of trichomes in *Woolly* dominant mutant turned into gland. This observation verifies that *GCR1* may not be involved in trichome initiation mediated by *Woolly* (see the images below).

3. Interestingly, our Y1H results showed *Woolly* indeed was able to interact with the promoter of *GCR1/2*. However, we still don't know the biological relevance of this interaction. It still needs further study to understand how the upstream master regulator affects the expression of a downstream negative regulator.

Reviewer #3 (Remarks to the Author):

The manuscript titled with “Spatiotemporal formation of glands in plants is modulated by MYB-like transcription factors” by Chang et al. demonstrates GCR MYB transcription factors interacted with and inhibited the gland-promoting regulator AP2/ERF transcriptional factor SITO1, and both of them interacted with transcriptional co-repressor SITPL2. By analyzing the published transcriptome data, the authors selected two uncharacterized MYB transcription factors, GCR1 and GCR2, which were predominantly expressed in the glandular trichomes. The *gcr1* single mutants exhibited the conversion of some non-glandular trichomes to glandular trichomes, and the double mutants *gcr1 gcr2* generated supernumerary glands, and frequently produced peltate trichomes with supernumerary glandular heads. GCR1 was expressed in tomato trichomes. The ectopic expression of GCR1/2 entirely inhibited gland formation, which was opposite to the phenotype of *gcr1 gcr2* mutant. The authors performed yeast two-hybrid screen for GCR1 interactors in using a cDNA library, and found that SITO1 interacted with GCR1. The interaction was confirmed by yeast two-hybrid, BiFC and pull-down assays. The overexpression of miR172-resistant-rSITO1 caused the increased glandular cells in digitate trichomes and peltate trichomes. The authors showed that SITO1 formed complex with

GCR1/2 and SITPL2 to directly repress GCR1/2. They further showed that GCR1/2 interacted with SITPL2 and directly repressed the expression of LFS is involved in the control of trichome formation.

The manuscript provides interesting results that showed two MYB transcription factors GCR1/2 redundantly promote glandular cell fate during tomato trichome formation. In this process, GCR1/2 act as repressors to inhibit LFS and itself by forming complexes with SITO1 and SITPL2. However, I think that the writing and data presentation have a lot of errors. And more data and experiments are required to support the working model and the conclusion of the manuscript. The detailed concerns were listed below.

Major concerns:

1. The introduction is so simple that many important information facilitating the understanding of the results are lack. In the introduction, the main information that most readers may not know should be provided. For examples, how many types of trichomes in the tomato? What is the main related progress in the molecular mechanisms underlying the formation of different trichomes? What is the EAR motif? What are the known functions of SITO1 and SITPL2, and etc.

Thanks for the valuable suggestion! We have re-wrote the manuscript and provided more detailed introduction of the study.

2. At least two alleles of gene mutants should be analyzed or the complementation should be provided to support that the loss of the gene function led to the defects of trichome formation.

Thank you! We added two alleles for *cr-gcr1*, *cr-gcr2*, *cr-gcr1/2* (MT), *cr-gcr1/2* (AC), *cr-gcr1/2* (LA1589), *cr-sito1* and *cr-lfs* created by CRISPR/Cas9 in the revised paper.

3. The authors described that “trichomes was replaced by ever-growing trichomes with supernumerary glands” “the short glandular trichomes (peltate trichomes, PT) often produced supernumerary glandular heads in double mutants”. How many? How often? I suggested that different kinds of trichomes but not only non-glandular trichomes and glandular trichomes should be statistically analyzed to resolve the question.

Thank you for the comment and we agree with the reviewer that the statistic quantification is needed for the phenotype description. In the revised manuscript, to quantitatively evaluate the glandular trichome phenotype, we divided all trichomes into three categories: NGT (non-glandular trichomes including type II, III and V), DGT (digital glandular trichomes including type I and IV) and PGT (peltate glandular trichomes including type VI and VII) (Fig. S1A)

In *gcr1/2* double mutant, 47 out of 52 type I trichomes are ever-growing trichomes, while 112 out of 122 type VI trichomes displayed supernumerary glandular heads.

4. What is the trichome phenotypes when the function of SITO1 is compromised. The ectopic expression of SITO1 could not validate the real function of SITO1 in the control of trichome formation.

To address the comment, we used CRISPR/Cas9 to create *cr-sltoe1* mutant. DGT is mainly existed at a juvenile phase and hardly found on the adult leaves and stems, so we analysis the trichomes phenotype of juvenile stems of WT and *cr-sltoe1* mutant. Compared with WT, *cr-sltoe1* mutant showed the reduction of DGT, while increased NGT, suggesting SITO1 is required for glandular trichome formation. However, not all glandular trichomes are suppressed in *cr-sltoe1* mutant, suggesting the potential functional redundancy exists. The phenotype and quantification data were present in Figure 3F and G.

5. The binding of GCR1 and GCR2 in the promoters of GCR1 and GCR2 should be tested using ChIP analysis in vivo.

As suggested by the reviewer, we performed ChIP-qPCR to test whether GCR1 binds to the promoter of its own and LFS using 35S:GCR1-GFP transgenic plants. The results support the conclusion drawn from Y1H and biotin-I experiments, and are now shown in Figure 4F and Figure 5F in the revised manuscript.

6. Does SITO1 bind to the promoters of GCR1 and GCR2?

We tested this possibility by Y1H. The 3Kb sequence upstream of GCR1/2 translation initiation site was cloned into pLacZi vector and SITO1 was cloned into pJG vector. Pairs of recombinant constructs were co-transformed into yeast strain EYG48. The interaction was detected on SD/-Ura/-Trp+X-gal medium. Our results showed no interaction between SITO1 and GCR1 /2 promoters.

7. Genetic interaction between TOE and GCR1 is necessary, what is the phenotype of double mutant *gcr1 sltoe1*?

We obtained *gcr1/sltoe1* double mutant by crossing *cr-gcr1* and *cr-sltoe1*. Phenotype analysis showed that phenotype of *gcr1/sltoe1* double mutant was the same as that of *gcr1* single mutant. We have added this result in Supplementary Figure11B and C.

8. GCR1 and GCR2 could directly interact with SITPL2 via its own EAR motifs. Why do GCR1 and GCR2 need to recruit SITPL2 by interacting with SITO1? Which domain of GCR1 and GCR2 is responsible for their interaction with SITO1? And which domain of SITO1 is responsible for its interaction with GCR1 and GCR2?

Thanks for bring this question! It is interesting that GCR1/2 can recruit SITPL2 directly by themselves, and can also by SITO1. According to our results, SITO1 can dramatically enhance GCR1/2 capacity to inhibit their own expression (Fig 3B). This makes sense if we combine the spatiotemporal expression of GCR1/2 and the genetic phenotypes. GCR1/2 expression is initiated at a certain level in the early stage of trichome development, and becomes gradually enhanced and restricted in the apical cells of the non-glandular trichomes. In the loss-of-function mutant of GCR1/2, we observed defects of cell division in addition to gland phenotype, suggesting the potential involvement of GCR1/2 in the early trichome cell division. In support of this speculation, pMTR1 promoter driven GCR1/2 showed inhibited trichome cell division.

Therefore, GCR1/2 level appears to be fine-tuned during trichome development. In the early trichome initiation when SITO1E1 is not expressed, GCR1/2 plays a weak self-inhibition role to maintain a certain level. When it comes to the differentiation into gland cells in the apical region, GCR1/2 need to be kept lower level and this requires the SITO1E1 activity. In line with this, SITO1E1 expression is initiated at the apical cells when the gland cells start to form (also see the figure below). We have discussed this in the revision manuscript.

To further address the interacting domain in GCR1/2 protein, we divided GCR1/2 into N-terminus part containing MYB-like domain, and C-terminus part containing EAR motif. We found GCR1/2 interacted with SITO1E1 via C-terminus part (Fig. 2, Fig. S8C). In addition, we examined the interacting domain in SITO1E1. To this end, we divided SITO1E1 into N-terminus part containing both EAR-like motif and AP2 domain, and C-terminus part containing EAR motif. The result showed that both domain could interact with GCR1/2 (Fig. S9).

9. “Furthermore, we found that the expression of GCR1/2 was strongly inhibited in pMTR1: rSITO1E1-GFP, but interestingly became significantly enhanced in *cr-gcr1/2*”. Does the repression of GCR1/2 by SITO1E1 require GCR1/2? How about the expression level of GCR1/2 in the loss of function mutant *slt1e1*?

Thank you for this great question! To test whether the repression of GCR1/2 by SITO1E1 requires GCR1/2, we conducted LUC reporter assay in *gcr1/2* double mutant protoplasts. The results showed that SITO1E1 alone could still inhibit GCR1/2 expression. However, such inhibition was significantly enhanced when GCR1/2 proteins were present. We also repeated this experiment in wild type tomato protoplasts and *gcr1/2* double mutant protoplasts for many times and the result are highly reproducible. Those results were showed in Figure S12D and E.

We think this regulation makes a lot of sense if we think of the developmental process of glandular cells. After the trichome initiation, the GCR1/2 started to express and the expression was gradually enhanced in the apical cells of developing trichome cells. When the glandular cells start to form, the primed apical cells start to express SITO1E1 and the expression was gradually enhanced in the apical glandular cells (see image below). In this way, the rising GCR1 expression needs to be quickly turned down, and the weekly expressed SITO1E1 needs a more efficient way to do this. In this stage, binding of SITO1E1 to GCR1 provides a more efficient way to reverse the rising GCR1 expression. Once this trend is reversed and the GCR1 level becomes lower than TOE1, SITO1E1 presumably can directly inhibit GCR1 expression and eventually

remove it entirely from the forming glandular cells. We have added this speculation in the discussion.

The expression level of SITO1 was upregulated in the *cr-sltoe1* juvenile leaves.

Figure: the expression level of GCR1/2 in the *cr-sltoe1* juvenile leaves.

10. The binding of GCR1 and GCR2 in the promoter of LFS should be also proved using ChIP analysis in vivo.

Please see the response to comment 5.

11. BiFC, LCI, or Co-IP should be used to verify that the protein interactions in vivo (like GCR1 and GCR2 interacting with TPL2, and SITO1 interacting with TPL2).

We performed BiFC and Co-IP to test the in vivo interaction between GCR1/2, TPL2 and SITO1. The results showed that both GCR1/2 and SITO1 interacted with TPL2, whereas mutations in the EAR motif of GCR1/2 or SITO1 disrupted the interaction. Together with results mentioned above, GCR1/2 and SITO1 appear to interact with TPL2 via EAR motif.

12. Fig.3B and Fig.5C, the control combinations including GCR1/2+rSITO1mEAR and GCR1mEAR+TPL2 should be included.

We have added the requested controls (now in Fig.4C and Fig.6C).

Minor concerns:

1. Line 49-51, No methods or description were found in “Materials and methods” in screening genes with the high expression level in the glandular trichomes. The authors could describe here and cited the published reference of RNA-seq data here.

Thanks for your suggestions. We have uploaded transcriptome data to the NGDC (<https://ngdc.cnpc.ac.cn/>). Accession numbers are CRA013731 (Transcriptome data of trichomes of WT, *cr-gcr1/2* and *pMTR1:GCR1*) and CRA011835 (Transcriptome data of *S.pennellii* trichomes and stems with removed trichomes). We also described the methods for screening the key genes in glandular trichomes and cited the published reference of RNA-seq data.

2. Line 58, GCR1/2 have three EAR motifs as shown in Fig. S1D, so here “motif” should be “motifs”.

We have revised it.

3. The different mutants of *gcr1/2* and the double mutants should be described in the text. What are the differences of phenotypes between them?

Thanks for the suggestion. *gcr1* single mutant caused the conversion of about 26% of trichomes into glandular trichomes, whereas *gcr2* single mutant had no significant trichome phenotype. In contrast, *gcr1/2* double mutant had dramatic phenotype and 96% of trichomes are glandular trichomes.

4. Line 66, “trichomes in double mutants became significantly elongated”. The length of the trichomes and the number of cells consisted of trichomes should be analyzed and statistically calculated to conclude that the difference is significant.

Thank you for the suggestion. We quantified the trichomes in both WT and *gcr1/2* double mutant. Type I trichomes (the longest type in tomato) of WT have average length of 3 mm, containing 8 cells along the trichome cell file. In *gcr1/2* mutant, the length of type I trichomes increased to about 5 mm (based on n=30 individual trichomes), containing about 30 cells along the trichome cell file. The quantification data presents Supplementary Figure 4.

5. Figure 1E, the control should be provided. That is, the image of glandular cell marker line in wild type and single mutants should be added. In the Fig. legend, “glandular cell maker line” has a typo. Maker should be marker?

We added the control (the staining of DGT (digital glandular trichomes, type I and type VI) and NGT (non-glandular trichomes, type II, type III and type V in WT) in Figure 1E. We also presented images of WT, single mutants and double mutant in Supplementary Figure 3.

6. Figure 1H, “with the expression stronger in apical cells of the non-glandular trichomes than glandular cells of glandular trichomes”. Which image in Fig. 1H is non-glandular trichome or glandular trichome? From the image, I found that in the middle image, the fluorescence of apical cells was also strong. How about the expression of GCR2? Here, the author did not give the data of time course, so in the figure legend, “Spatio-temporal expression of GCR1” is not accurate.

Thanks for the comment. Different type trichomes were indicated in the Figure 1H. Besides, Propidium Iodide (PI) staining was used to present the cell edge. We have moved the previous images and quantification data to Supplementary Figure 5A because of its poor PI staining and added another line to Figure 1H. In Figure 1H, we quantified the fluorescence intensity of NGT, PGT and DGT.

We have added the expression of GCR2 in supplementary Figure 5B. In line with the single mutant phenotype, GCR2 showed very low expression, which makes the precise fluorescence quantitation difficult.

7. Figure 1I, the trichome phenotypes on the hypocotyl should be also statistically analyzed. Since in Fig.1A appear to present the trichome defects in the leaves (the authors should be provided the organ information in text and figure legend), the phenotype of trichomes in leaves of pMTR1:GCR1-GFP should be also statistically analyzed.

In Figure 1I, the trichome phenotypes on the hypocotyl were statistically analyzed (now in Figure 1J). Fig. 1A presents the phenotype of *cr-gcr1/2* on the stems and we also quantified it. We have provided the organ information in text and figure legend.

8. Line 90-91, “yeast two-hybrid experiments showed that GCR1/2 mainly interacted with SITO1 through the C-terminus”, in the Fig. 2. Legend “Y2H assay. GCR1/2 interacted with SITO1 through its N-terminal domain.” From the Fig 2A, the interaction domain should be C-terminus.

We have corrected it.

9. I suggested that the authors indicate the different kinds of trichomes in the phenotypic images of Figures.

Thanks for your suggestions. Different type trichomes were indicated in Figure 1.

10. Line 93, “Spatiotemporal expression analysis of pSITO1: GUS revealed that SITO1 was highly expressed in the glandular cells of glandular trichomes.” No time course GUS staining were shown, “Spatiotemporal expression” is not accurate.

We have changed “Spatiotemporal expression” to “Expression pattern”.

11. Line 94 “SITO1” and many gene names in the manuscript are not italic. Gene names should be italic.

We have corrected it.

12. Many abbreviations have no full names. The abbreviations used at the first time should be given full names in the Abstract and the main text of the manuscript.

We have added.

3. Figure 2E, the phenotype in the image should be statistically analyzed.

We have quantified the trichomes.

14. Line 129-130, “SITO1 acts as the transcriptional repressor via the interaction between its bearing EAR motif and TPL in Arabidopsis”. Here, SITO1 should be TO1 in Arabidopsis?

Yes, we have revised it.

15. The Figure legends for Fig.3B and Fig.5D are missed.

We have added.

16. In Fig.4G, the phenotype of gcr1/2 and lfs should be included as the controls.

We have added.

17. The Discussion is not sufficient.

We have re-wrote the discussion.

18. Fig 1A-D lacks the scale bar for the enlarged images, and Fig 1F lacks significance analysis.

We have added them.

19. Fig 2E lacks significance analysis. How many biological replicates were performed?

We have added the significance analysis. In total, we obtained at least 30 lines and six lines were used for analysis the transcription level and phenotype.

20. Fig 2D legend misses the unit after the bar. (line 590, “Bar: 25.”)

We have added it.

21. How many transgenic lines were obtained for GUS analysis in Fig 2D?

We have obtained two pSITOE1: GUS lines. To quantitation the transcriptional level of SITOE1, we constructed another reporter line of pSITOE1:NLS-Venus. At least 10 independent lines were used for analysis (Fig 3A).

22. The three replicates of WT should be put together in Fig S3B, Fig S9, and the significant analysis should be added.

Thanks for your suggestions. We have revised it.

23. The significant analysis should be performed in Fig 3A.

We have added it.

24. “Figure S9. Genotype and phenotype of *gcr1/gcr2/lfs* triple mutant”. No data about the triple mutants was found in the Fig. S9.

Sorry for inaccurate figure legend. Phenotype of *gcr1/gcr2/lfs* triple mutant was showed in Fig 5H. *gcr1/gcr2/lfs* triple mutant was obtained by crossing *cr-gcr1/2* and *cr-lfs*, so genotype of *gcr1/gcr2/lfs* triple mutant is the same as *gcr1/2* double mutant and *lfs* single mutant.

Reviewer #4 (Remarks to the Author):

This manuscript investigates a transcription factor regulatory network in tomato that is involved in the differentiation of glands from capitate trichomes (type IV). Two MYB TFs (GCR1/2) were found as repressors of the gland cell differentiation. They are antagonized by another TF (SITOE1) which represses their expression and they repress LFS, which is a promoter of gland formation.

The data presented is of high interest for people working on tomato trichomes, and provide insight as to why cultivated tomato has few capitate trichomes in adult leaves. However, the quality of the execution and presentation of the data as well as the quality of the writing need much improvement. This makes for difficult reading. Figure legends are often incomplete or imprecise.

Major Comments:

1) The high expression of GCR1 and GCR2 in wild tomatoes (*S. pennellii*) contradicts the role of GCR1 and GCR2 as repressors of type IV trichomes, because *S. pennellii* has extremely high density of type IV trichomes, as the authors rightly point out. This is certainly concerning and I am wondering if the authors have an explanation for this.

Thanks for the question! We are sorry for not describing this clearly. The high expression in wild tomatoes (*S. pennellii*) was actually compared with the leaves where GCR1/2 almost had no expression. We compared the transcriptome between extracted trichomes and stem epidermis with all trichomes removed.

To make the expression pattern of GCR1/2 clearer, we made PGCR1:NLS-Stadygold line. Confocal visualization indicates that GCR1 can express in most trichomes with

low level, but is highly enriched in the non-glandular apical cells and has extremely low expression in glandular heads.

To further clarify this question, we introduced *pMTR1:GCR1-GFP* into the wild tomato species *S.pennellii* LA0716 by stable transformation. The result showed that ectopic expression of *GCR1* inhibited glandular cell formation. This result was shown in Fig. S7E of the revised manuscript.

2) What is the pMTR1 promoter? I could not find any information on it in the text.

Sorry for not making this clear in the last submission. Promoter of MTR1 was previously reported by our group (Wu, et al., Dev Cell, 2023), which is specifically active in tomato trichomes. In particular, pMTR1 activity is higher in apical cells of tomato trichomes, which makes it an ideal promoter to drive the specific expression of a gene in apical trichome cells. In the revised manuscript, we added the expression pattern of pMTR1 in Fig. S6A.

3) Line 158-162: the interpretation is not consistent with the genotypes of the *lfs* mutants. According to Fig. S9, both mutants are bi-allelic with mutations that result in the same aberrant proteins (in each mutant). Therefore no phenotypic difference would be expected between those two mutants. This brings another question: are the plants characterized the first generation transformants or the next generation? If they are from the next generation, then segregation can occur, but no simple heterozygote will be produced, only homozygous for each individual mutation or double heterozygote (like the parent).

We used the next generation to detect the phenotype of *lfs* mutant. However, as the *lfs* null mutant fail to produce cotyledons and leaves (Capua, Y., 2017, PNAS), we only selected a heterozygote (line #1) or double heterozygote mutant (line #3) to analysis the phenotype. This has also been described in our previous article (Wu., 2023, Developmental cell).

In Fig. S16A, one heterozygote (line #1, with 50% of edited DNA strands) and one double heterozygote mutant (line #3) were present. Therefore, their phenotypes are different.

4) Since type IV trichomes produce acyl sugars, it would be essential to measure these in all mutants characterized here.

Using the previously published method by Dr. Rob Last group (Yann-Ru Lou, et al., 2019; Leong et al., 2019; Ghosh et al., 2014), we measured the acyl sugar levels in *cr-gcr1/2*, *PMTR1:GCR1*, *PMTR1:GCR2*, *PMTR1:SITOE1* and *cr-lfs* lines. The results are in Supplementary Figure 20 of the revised manuscript.

References:

Yann-Ru Lou, Bryan Leong, 2019. Leaf surface acylsugar extraction and LC-MS profiling - v1.0. protocols.io <https://dx.doi.org/10.17504/protocols.io.xj2fkqe>.

Leong BJ, Lybrand DB, Lou Y, Fan P, Schillmiller AL, Last RL, 2019. Evolution of metabolic novelty: A trichome-expressed invertase creates specialized metabolic diversity in wild tomato. Science Advances 5(4). doi:10.1126/sciadv.aaw3754

Ghosh, B., Westbrook, T. C. & Jones, A. D. Comparative structural profiling of trichome specialized metabolites in tomato (*Solanumly copersicum*) and *S.*

habrochaites: acylsugar profiles revealed by UHPLC/MS and NMR. *Metabolomics* 10(3), 496–507 (2014).

5) An accession number for the transcriptome data produced in this manuscript must be provided.

We have uploaded transcriptome data to the NGDC (<https://ngdc.cnbc.ac.cn/>). Accession numbers are CRA013731 (Transcriptome data of trichomes of WT, *cr-gcr1/2* and *pMTR1:GCR1-GFP*) and CRA011835 (Transcriptome data of *S.pennellii* trichomes and stems with removed trichomes).

6) There is a number of issues with the figures and presentation of the data:

a. Figure 1: as there are several types of glandular trichomes (type VI, type I/IV and type VII, the latter could be ignored). A much more specific counting of the different types is required. The authors should adopt the accepted nomenclature for the different types of trichomes (type I until VII).

We have performed the quantification of the trichomes shown in Fig. 1I and Fig. S3F. We quantified all seven type trichomes. To quantitatively evaluate the glandular trichome phenotype, we divided all trichomes into three categories: NGT (non-glandular trichomes including type II, III and V), DGT (digital glandular trichomes including type I and IV) and PGT (peltate glandular trichomes including type VI and VII) (Fig. S1A). Also based on the comment from reviewer 1, we combined the Fig 1I and Fig. S3F and all the results are now in Fig. 1J in the revised paper.

b. Fig. 1: in panel H the bright field image should be provided. GCR1 appears to be expressed not just in the tip cell but in the stalk cells of the trichomes as well and in type VI glandular trichomes. Which raises the question, why when GCR1 is overexpressed it should affect type VI trichomes in cotyledons as shown in panel I?

In panel H, we employed the Propidium Iodide (PI) staining which gives rise to the clear cell edge. We have moved the previous images and quantification data to Fig. S 5A because of its poor PI staining and added another lines to Fig. 1H.

GCR1 is expressed in all trichome cells and highly expressed in the tip cells of non-glandular trichomes. When GCR1 is overexpressed, all type VI trichomes are affected not only in cotyledons.

c. Figure 2 (and others): Please provide complete western blots and coomassie gels with visible size markers for all IP and pull down experiments.

We have added. We also added the original figures at the end of this file.

d. Fig. 2: provide trichome counts for panel E

We have added.

e. Fig. 3: was the pull down done with purified GCR1? A coomassie gel should be shown. If not purified, a control with extract from a strain with the empty vector should be made.

Biotin-IP was done with purified His-GCR1 and His-GCR2 in Fig 4E. We have added a GST-HIS protein as a negative control. Besides, we also verified the promoter fragment of GCR1/2 with mutant motif could not interact with GCR1/2 protein.

f. Fig. 3: there is a problem with the legend. Panel B is missing, legends of panels have been mixed up.

We have corrected it.

g. Figure 4: again provide trichome counts. And same remark as in Figure 3 for the DNA IP assays.

We have quantified trichomes of WT, *cr-gcr1/2*, *cr-lfs* and *cr-gcr1/2/cr-lfs*. Purified His-GCR1 and His-GCR1 were used for DNA IP assays. We have added a GST-HIS protein as a negative control. Besides, we also verified the promoter fragment of LFS with mutant motif could not interact with GCR1/2 protein..

h. Figure 5: Panel E is not explained. Where are the *lfs* mutants? What is *cr-gcrs*? There is something missing in the legend.

We have explained Panel E and changed “*cr-gcrs*” to *cr-gcr1/2*.

i. Fig. S2: The mutations in GCR genes in the different tomato backgrounds (panel D) must be provided.

We have added the sequence analysis of GCR genes in AC and *S. pimpine* LA1589 in Fig. S7.

j. Fig. S3: In panel A *pMTR1* confers expression in the whole trichome. Why is it only in the tip cell when fused to GCR1 (panel C). Again, provide trichome counts.

pMTR1 expressed in the whole trichome and highly expressed in the tip cell. GCR1 fused with *pMTR1* also expressed in all trichome cells.

k. Please provide source organisms for NbGCR and PeGCR1.

Hypocotyls of *pMTR1: NbGCR* and *pMTR1: PeGCR* were showed. We have added in the figure legend. Figures and Figure legends are now presented in Figure 1.

l. Fig. S3. The lines characterized here are not the same as in Fig. 3. when they should be.

Yes, you are correct. In Fig. S3, we presented the *pMTR1:GCR-GFP* line, while in Fig. 3A, we showed the F1 line of the cross between *pMTR1:GCR1-GFP* and *pMTR1:SITOE1-GFP*.

m. Fig. S7: provide full western blots with size markers visible and coomassie with size markers clearly labelled.

We have added it. We also added the original figures at the end of this file.

n. Fig. S8: trichome counts are need. Was the expression determined from whole leaves or just trichomes (panel E). Panel F: the value of the scale bar cannot be right.

We have quantified the trichomes. The expression determined from young leaves. The the value of the scale bar was corrected.

o. Fig. S9: see comment 2) on the interpretation of the *lfs* mutants. And the legend says “phenotype of *gcr1/gcr2/lfs* triple mutant”. I only see *lfs* single mutants.

Sorry for incorrect figure legend. In fig. S9, only *lfs* mutants was showed. *gcr1/gcr2/lfs* triple mutant was present in Fig 5H.

p. Fig. S9: Microtom has hardly any type IV trichomes on adult leaves. Are the pictures from cotyledons?

Hypocotyl of *cr-lfs* was shown in Fig. S16C.

Other comments:

1) is miR172 present in tomato? Corresponding references should be cited.

Yes, tomato has miR172, which has been reported by many previous literatures (Zhang et al., 2008; Chung et al., 2020; Lin et al., 2021)

References:

Lin W, Gupta SK, Arazi T, et al. MIR172d is required for floral organ identity and number in tomato. *International Journal of Molecular Sciences*, 2021, 22(9):4659. DOI:10.3390/ijms22094659.

Chung MY, Nath UK, Vrebalov J, et al. Ectopic expression of miRNA172 in tomato (*Solanum lycopersicum*) reveals novel function in fruit development through regulation of an AP2 transcription factor. *BMC Plant Biology*, 2020, 20 (1). DOI:10.1186/s12870-020-02489-y.

Zhang J, Zeng R, Chen J, et al. Identification of conserved microRNAs and their targets from *Solanum lycopersicum* Mill. *Gene*, 2008, 423 (1):1-7. DOI:10.1016/j.gene.2008.05.023.

2) Line 200: why should the repression via TPL2 be epigenetic? I did not see any evidence for this in the manuscript.

TOPLESS (TPL) has been reported in many plant species as a transcriptional repressor. The mechanism of this protein function is mainly through histone deacetylases. This role has been widely described (Long et al., 2006; Kagale et al., 2011; Krogan et al., 2012, Deng et al., 2022). In the revised manuscript, we turned down the conclusion and made it clear that we speculate the epigenetic repression by TPL2 based on the previous studies.

References:

Long, J. A., Ohno, C., Smith, Z. R. & Meyerowitz, E. M. TOPLESS regulates apical embryonic fate in *Arabidopsis*. *Science* 312(5779), 1520–1523 (2006).

Kagale, S. & Rozwadowski, K. EAR motif-mediated transcriptional repression in plants: an underlying mechanism for epigenetic regulation of gene expression. *Epigenetics* 6(2), 141–146 (2011).

Krogan, N. T., Hogan, K. & Long, J. A. APETALA2 negatively regulates multiple floral organ identity genes in *Arabidopsis* by recruiting the co-repressor TOPLESS and the histone deacetylase HDA19. *Development* 139(22), 4180–4190 (2012).

Deng, H. et al. SIERF.F12 modulates the transition to ripening in tomato fruit by recruiting the co-repressor TOPLESS and histone deacetylases to repress key ripening genes. *Plant Cell* 34(4), 1250–1272 (2022).

Fig. 4E Biotin-labeled DNA IP assays show GCR1 interacts with *pGCR1-1* and *pGCR1-2* and GCR2 interacts with *pGCR2-1* and *pGCR2-2*

Fig. 5G Biotin-labeled DNA IP assays show GCR1 interacts with *pLFS-2* and *pLFS-5* and GCR2 interacts with *pLFS-4* and *pLFS-5*.

Fig. 2A Co-IP assay shows GCR1/2 and GCR1/2-C could interact with SITO1

Original figures

Fig. 6A CoIP shows GCR1 and GCR2 interact with TPL2 by EAR motif.

Original figures

Fig. S14 Co-IP assay shows that SITO1 interacts with TPL2

Original figures

Fig. S8D GST-pull down assay show that GST-SITOE1 but not GST pulls down HIS-GCR1 and HIS-GCR2.

(Previous Fig. 2C)

Fig. S14B GST-pull down. GST-TPL2 but not GST pulls down His-SITOE1.

(Previous Fig. S7B)

Fig. S17C GST-pull down assay show GCR1/2 interact with TPL2. (We re-performed this experiment.)

Original figures

REVIEWERS' COMMENTS

Reviewer #1 (Remarks to the Author):

The authors have nicely addressed my questions. The manuscript has been significantly improved.

Reviewer #2 (Remarks to the Author):

The manuscript is very significantly improved compared with the previous section. While there are a few additional issues to address (see below), from my perspective, they are rather minor.

Since the authors used Arabidopsis as an example for mechanisms governing trichome formation (I personally think that Arabidopsis is an exception, rather than the rule), I wonder to what extent the results show in this manuscript shed some light on mechanisms regulating lateral inhibition of trichome initiation. For example, are the two MYBs functioning in a cell autonomous fashion?

Comments

1. Change secondary metabolite to specialized metabolite, as it is now accepted in the field
2. Lines 184-186: More details are needed for the Biotin-IP and CHIP-PCR experiments – antibody? Native or overexpression?
3. Lines 465 – 467: A better description of the statistical tests needs to be provided not only in the Methods, but in each figure legend
4. The Data Availability statement is meaningless as written
5. English language issues persist – a few examples are listed (but there are many others):
 - a. Line 43: of important should read of importance
 - b. Line 56: morphology should read morphologies
 - c. Line 98: remove the “probably” as you already say “suggesting”
 - d. Line 105: is glandular cell correct? Perhaps glandular structures is more appropriate?
 - e. Line 112: should be gland repressors
 - f. Line 118: GCR1 is expressed?

g. Line 193: a strong interaction

h. Line 579-580: length of between – use one or the other

Reviewer #3 (Remarks to the Author):

The revised manuscript has been highly improved and most of my concerns have been resolved. However, I still have some concerns about the Introduction and the manuscript writing. Some are listed below.

1. In introduction, the authors provided some more backgrounds. I think the introduction is still too simple. The EAR motif, the known functions of SITO1 and SITPL2, and more background knowledge about the trichome fate determination or differentiation should be added.
2. Line 133, does TOE1 indicate Arabidopsis TOE1 or SITO1? I suggested that the authors added SI when they described genes or proteins from tomato. The authors sometimes added, and sometimes did not in the whole manuscript.
3. Line 167, “gcr1slt1” should be “gcr1 slt1”.
4. Line 179-181, How do “GCR homologous genes” bind to DNA?
5. Line 192, “6 TPL genes”, TPL should be italic.
6. Line 265 “forced expression of GCR genes”, GCR should be italic. Please check this type of typos in whole manuscript.
7. Line 311 what is the “the 3K upstream of” meaning?

REVIEWERS' COMMENTS

Reviewer #1 (Remarks to the Author):

The authors have nicely addressed my questions. The manuscript has been significantly improved.

Reviewer #2 (Remarks to the Author):

The manuscript is very significantly improved compared with the previous section. While there are a few additional issues to address (see below), from my perspective, they are rather minor.

Since the authors used Arabidopsis as an example for mechanisms governing trichome formation (I personally think that Arabidopsis is an exception, rather than the rule), I wonder to what extent the results show in this manuscript shed some light on mechanisms regulating lateral inhibition of trichome initiation. For example, are the two MYBs functioning in a cell autonomous fashion?

Thanks for the thoughtful comment! It is possible a lateral inhibition mechanism plays a role in regulating tomato trichome initiation. Currently we have unpublished data showing another protein, MTR1 is possibly involved in the putative lateral inhibition in tomato. However, we found no evidence showing the non-cell-autonomous role for the two MYBs we reported in this study.

Comments

1. Change secondary metabolite to specialized metabolite, as it is now accepted in the field

We have changed secondary metabolite to specialized metabolite in the revised manuscript.

2. Lines 184-186: More details are needed for the Biotin-IP and ChIP-PCR experiments – antibody? Native or overexpression?

We have added the detailed information for these experiments in the revised manuscript.

3. Lines 465 – 467: A better description of the statistical tests needs to be provided not only in the Methods, but in each figure legend

Thanks for your suggestion! We have provided more details of the statistical tests in the revised figures and figure legends.

4. The Data Availability statement is meaningless as written

We have rewritten this section.

5. English language issues persist – a few examples are listed (but there are many others):

- a. Line 43: of important should read of importance
- b. Line 56: morphology should read morphologies
- c. Line 98: remove the “probably” as you already say “suggesting”
- d. Line 105: is glandular cell correct? Perhaps glandular structures is more appropriate?
- e. Line 112: should be gland repressors
- f. Line 118: GCR1 is expressed?
- g. Line 193: a strong interaction
- h. Line 579-580: length of between – use one or the other

We have revised these accordingly and thoroughly checked the other parts of the manuscript.

Reviewer #3 (Remarks to the Author):

The revised manuscript has been highly improved and most of my concerns have been resolved. However, I still have some concerns about the Introduction and the manuscript writing. Some are listed below.

1. In introduction, the authors provided some more backgrounds. I think the introduction is still too simple. The EAR motif, the known functions of SITO1 and SITPL2, and more background knowledge about the trichome fate determination or differentiation should be added.

We have added more background about the trichome fate determination or differentiation in Introduction. In addition, we discussed more about EAR motif and the known functions of SITO1 and SITPL2 in the Discussion section.

2. Line 133, does TOE1 indicate Arabidopsis TOE1 or SITO1? I suggested that the authors added SI when they described genes or proteins from tomato. The authors sometimes added, and sometimes did not in the whole manuscript.

We have checked the whole manuscript and added all SI when necessary.

3. Line 167, “gcr1s1toe1” should be “gcr1 sltoe1”.

Revised.

4. Line 179-181, How do “GCR homologous genes” bind to DNA?

We have changed GCR homologous genes to GCR homologous proteins.

5. Line 192, “6 TPL genes”, TPL should be italic.

Revised.

6. Line 265 “forced expression of GCR genes”, GCR should be italic. Please check

this type of typos in whole manuscript.

Revised.

7. Line 311 what is the “the 3K upstream of” meaning?

Sorry for the typo. It refers to the 3000 bp promoter region of GCR1/2. We have revised it.